Citation: *Molecular Systems Biology* 9:699
www.molecularsystemsbiology.com

# Sequential induction of auxin efflux and influx carriers regulates lateral root emergence

Benjamin Péret[1,2,3,12], Alistair M Middleton[1,2,4,12,13], Andrew P French[1,2], Antoine Larrieu[1,2], Anthony Bishopp[1,2,5], Maria Njo[6,7], Darren M Wells[1,2], Silvana Porco[1,2], Nathan Mellor[1,2], Leah R Band[1,2,4], Ilda Casimiro[8], Jürgen Kleine-Vehn[6,7], Steffen Vanneste[6,7], Ilkka Sairanen[9], Romain Mallet[1,2], Göran Sandberg[10], Karin Ljung[9], Tom Beeckman[6,7], Eva Benkova[6,7], Jiří Friml[6,7], Eric Kramer[11], John R King[1,4], Ive De Smet[2,6,7], Tony Pridmore[1], Markus Owen[1,4] and Malcolm J Bennett[1,2,*]

[1] Centre for Plant Integrative Biology, University of Nottingham, Loughborough, UK, [2] Division of Plant and Crop Sciences, School of Biosciences, University of Nottingham, Loughborough, UK, [3] Unité Mixte de Recherche 7265, Commissariat à l'Energie Atomique et aux Energies Alternatives, Centre National de la Recherche Scientifique, Aix-Marseille Université, Laboratoire de Biologie du Développement des Plantes, Saint-Paul-lez-Durance, France, [4] Centre for Mathematical Medicine and Biology, School of Mathematical Sciences, University of Nottingham, Nottingham, UK, [5] Department of Biosciences, Institute of Biotechnology, University of Helsinki, Helsinki, Finland, [6] Department of Plant Systems Biology, Flanders Institute for Biotechnology, Ghent, Belgium, [7] Department of Plant Biotechnology and Genetics, Ghent University, Ghent, Belgium, [8] Universidad de Extremadura, Facultad de Ciencias, Badajoz, Spain, [9] Department of Forest Genetics and Plant Physiology, Umeå Plant Science Centre, Swedish University of Agricultural Sciences, Umeå, Sweden, [10] Department of Plant Physiology, Umeå Plant Science Centre, Umeå University, Umeå, Sweden and [11] Physics Department, Simon's Rock College, Great Barrington, MA, USA

[12]These authors contributed equally to this work.
[13]Present address: University of Heidelberg, Im Neuenheimer Feld 267, 69120 Heidelberg, Germany.
* Corresponding author. Centre for Plant Integrative Biology, University of Nottingham, Sutton Bonington Campus, Loughborough, Leics LE12 5RD, UK.
Tel.: +44 115 951 3255; Fax: +44 115 951 6334; E-mail: malcolm.bennett@nottingham.ac.uk

In *Arabidopsis*, lateral roots originate from pericycle cells deep within the primary root. New lateral root primordia (LRP) have to emerge through several overlaying tissues. Here, we report that auxin produced in new LRP is transported towards the outer tissues where it triggers cell separation by inducing both the auxin influx carrier LAX3 and cell-wall enzymes. *LAX3* is expressed in just two cell files overlaying new LRP. To understand how this striking pattern of *LAX3* expression is regulated, we developed a mathematical model that captures the network regulating its expression and auxin transport within realistic three-dimensional cell and tissue geometries. Our model revealed that, for the LAX3 spatial expression to be robust to natural variations in root tissue geometry, an efflux carrier is required—later identified to be PIN3. To prevent LAX3 from being transiently expressed in multiple cell files, *PIN3* and *LAX3* must be induced consecutively, which we later demonstrated to be the case. Our study exemplifies how mathematical models can be used to direct experiments to elucidate complex developmental processes.
*Molecular Systems Biology* **9**: 699; published online 22 October 2013; doi:10.1038/msb.2013.43
*Subject Categories:* simulation and data analysis; plant biology
*Keywords:* 3D modelling; auxin transport; lateral root emergence; ODE

## Introduction

Plant root systems have a major role in nutrient and water acquisition from the soil and also provide anchorage (Smith and De Smet, 2012). The establishment of a branched root system relies on *de novo* formation of new organs, termed as lateral root primordia (LRP). In the model plant *Arabidopsis thaliana*, lateral roots (LRs) originate from pericycle cells located deep within the parental root overlaying the xylem pole (Figure 1A). Pairs of these xylem-pole pericycle (XPP) cells (termed as LR founder cells) undergo anticlinal (stages 0–I) and then periclinal and tangential divisions (stages I–II) to create dome-shaped primordia (stages III–VII) (Figure 1B) as defined previously (Malamy and Benfey, 1997; Lucas *et al*, 2013). As a result, new LRP have to emerge through overlaying endodermal, cortical and epidermal tissues (summarised in Figure 1B).

The mechanisms facilitating the emergence of LRP have puzzled scientists for over a century (reviewed in Péret *et al*, 2009). The auxin influx transporter LAX3 has recently been demonstrated to be important for LRP emergence in *Arabidopsis* (Swarup *et al*, 2008). *LAX3* exhibits a striking pattern of expression in just two files of cortical cells overlaying the new LRP (Figure 1C and D) that later undergo cell separation to facilitate organ emergence (Swarup *et al*, 2008). Auxin acts as a key signal that coordinates primordium outgrowth, outer tissue deformation and cell separation (Benková *et al*, 2003; Swarup *et al*, 2008; Lucas *et al*, 2013). We hypothesise that auxin does this by being transported from newly initiated LRP towards cells in overlaying tissues, where it induces genes such as *LAX3* that promote cell separation.

LAX3 controls the auxin-dependent induction of a set of cell-wall remodelling enzymes, including polygalacturonase (PG),

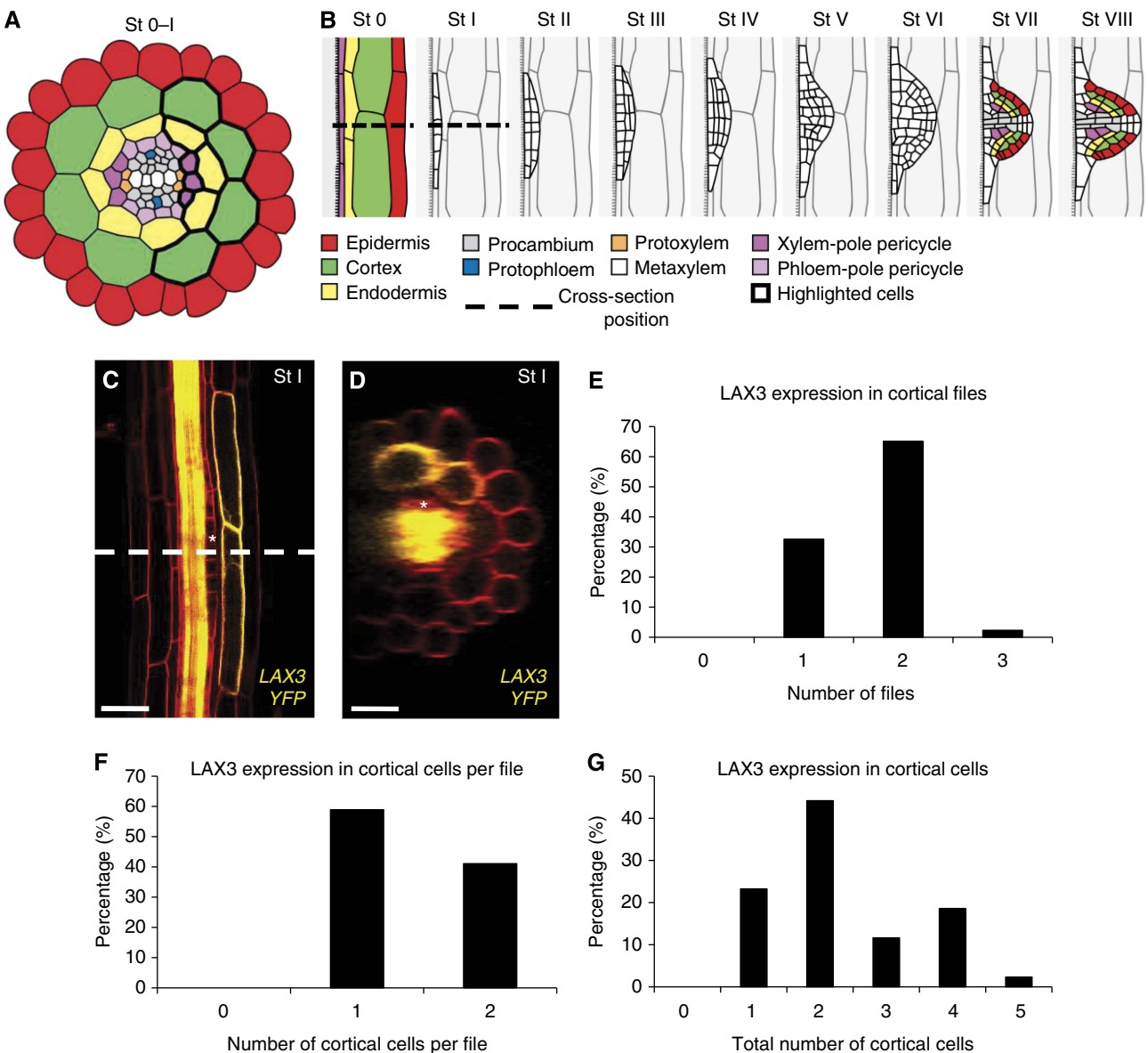

**Figure 1** Lateral root (LR) formation and emergence in *Arabidopsis thaliana*. (**A**) Cross-section of an *Arabidopsis* root (during stages 0–I of LRP emergence) showing the different cell types, with the position of the cross-section shown in (**B**). Xylem-pole pericycle cells are grouped in three cell files and are in contact with several endodermal cell files, which in turn about several cortical cell files (highlighted cells). (**B**) Stages of LR formation. Between stages 0 and I, the XPP cells (from which the LRP originate) undergo several rounds of anticlinal division. Note that in the transverse direction cells vary in length and appear in a staggered formation. (**C**, **D**) LAX3 protein accumulation pattern was visualised using a functional *pLAX3:LAX3YFP* fusion in a tangential root section (**C**) or a cross-section (**D**) (with the position of the cross-section shown indicated by the dashed line in (**C**)). The asterisk highlights the position of one of the XPP cells, from which the LRP originate. (**E–G**) Statistical analysis was performed on lateral root primordia of plants bearing a functional *pLAX3:LAX3YFP* fusion (*n* = 40) to determine the number of files (**E**), the number of cortical cells per file (**F**) and the total number of cortical cells (**G**) showing *LAX3* expression.

in cells overlaying LRP (Swarup *et al*, 2008). As a result of their enzymatic action, the walls of overlaying cells are weakened. Separation of these overlaying cells can be further promoted when LRP cells start dividing and expanding. *LAX3* is induced very early on during the LRP formation process, as its expression is observed from stage I onward, before any major changes in the morphology of the new LRP and overlaying tissues. The early induction of *LAX3* is likely to be required to synthesise and traffic cell-wall modifying enzymes to ensure that overlaying cells are ready to separate when new LRP start to protrude into outer root tissues (Figure 1B).

Several other components of the LRP emergence machinery have also been identified (Swarup *et al*, 2008). These include the transcription factors ARF7 and IAA14 that have key roles during LR formation (Fukaki *et al*, 2002; Okushima *et al*, 2005, 2007) and regulate auxin-inducible *LAX3* induction (Swarup *et al*, 2008). Auxin induces *LAX3* expression by mediating TIR1/AFB-dependent degradation of the transcriptional repressor protein IAA14, thereby releasing its interacting transcription factor ARF7 to trigger expression of downstream target genes such as *LAX3*.

Despite our detailed knowledge about the regulatory components that control auxin-inducible *LAX3* expression in

cortical cells overlaying new LRP, the molecular and tissue-scale mechanisms controlling its highly specific expression pattern remain unclear. In this study, we initially demonstrate that new LRP are able to channel auxin to overlaying cortical cells and induce *LAX3* expression. We then develop a mathematical model of the regulatory network controlling *LAX3* induction and couple it to one for auxin movement in a realistic 3D multicellular geometry. Our modelling efforts enable us to unravel the mechanisms regulating the influx carrier's spatial expression pattern. In particular, an iterative cycle of modelling and experimental perturbations revealed the existence of a new regulatory component, the auxin efflux carrier PIN3. A summary of the different model versions is provided in Table I. We test how robust the model is to natural variations in tissue geometry and the auxin source, and conclude that PIN3 has a key role. Finally, we predict that the *LAX3* expression pattern requires the sequential induction of auxin efflux and influx transporters, which we later demonstrate to be the case. Together, our results suggest that the localisation of the auxin source, together with sequential induction of *PIN3* and *LAX3*, can lead to sharp LAX3 expression patterns that are robust to variations in both tissue geometry and magnitude of the auxin source.

## Results

### *LAX3* is expressed in a limited number of cortical cells facing the LRP

The auxin transporter LAX3 displays a highly distinctive spatial expression pattern during LRP emergence. A functional *pLAX3:LAX3-YFP* transgene reveals that the LAX3 protein is specifically expressed in cortical cells overlaying new LRP (Figure 1C and D). In all, 65.1% of the LRP showed expression of *LAX3* in two cortical cell files versus 32.6% in one file and 2.3% in three files (Figure 1E). The number of cortical cells per file expressing *LAX3* was one (58.9% of LRP) or two (41.1% of LRP) (Figure 1F). As a result, the total number of *LAX3*-expressing cortical cells in the different cell files was between one and five (with the highest number only representing 2.3% of LRP; Figure 1G). Hence, typically LRP would induce *LAX3* expression in two cortical cell files each bearing one to two cortical cells, generating a total number of two to four cortical cells expressing *LAX3* (Figure 1C and D). This pattern is essential if cell separation is limited to occur between just two

cell files, which minimises damage to overlaying tissues that protect inner root tissues from soil pathogens. The establishment of such a highly specific expression pattern in front of a developing LRP raises the question of how this is achieved.

### The lateral root primordium acts as a source of auxin during organ emergence

*LAX3* is an auxin-inducible gene (Swarup *et al*, 2008). As a strong auxin gradient is established in the LRP, with its maximum at the apex (Benková *et al*, 2003), we hypothesise that this signal is channelled towards overlaying cortical cells where it induces *LAX3* expression. We tested whether new LRP provide the source of the auxin signal that induces *LAX3* expression (and the downstream effector genes such as *PG*) in overlaying cells (Swarup *et al*, 2008). We initially employed a genetic approach using the *pin2* mutant to examine the impact of elevating auxin accumulation at the LRP apex (Swarup *et al*, 2008). Quantitative RT–PCR shows that the mRNA abundance of both *LAX3* and *PG* was elevated in the *pin2* mutant background (Figure 2A–C). In addition, the expression of both p*LAX3:GUS* and p*PG:GUS* reporters was stronger in cells overlaying LRP in the *pin2* mutant background (Figure 2D–G). Hence, increased auxin accumulation in *pin2* LRP can be correlated with elevated levels of *LAX3* and *PG* in the outer tissue, suggesting that auxin can move from the LRP towards the outer tissue.

To induce *LAX3* expression in cortical cells overlaying new LRP as early as stage I (Figure 1C and D; Swarup *et al*, 2008), auxin must be able to move from the XPP cells as soon as they are mitotically activated (and so its initial expression pattern is established between stages 0 and I, see Figure 1B). To determine whether this is the case, we engineered all XPP cell files with the ability to synthesise auxin. This was achieved by expressing bacterial *iaaH-RFP* (indole-3-acetamide hydrolase and Red Fluorescent fusion protein) that converts an inactive auxin precursor indole-3-acetamide (IAM) into the bioactive form indole-3-acetic acid (IAA) (Blilou *et al*, 2005) and promotes LR development (Figure 3A and B). The *iaaH-RFP* gene was expressed in three files of XPP cells at opposite ends of the protoxylem using the J0121 driver line (Figure 3C–E and I). MS/MS measurements of *J0121»iaaH-RFP* roots treated with the *iaaH* substrate $^2H_5$-indole-3-acetamide (D5-IAM) confirmed that this auxin precursor was efficiently converted into D5-IAA (but not in the J0121 control samples; Figure 3F) resulting in increased expression of LAX3 and PG (Figure 3G

**Table I** Summary of different model versions and relevant figure references

| Model version | *LAX3* regulation | *AEC/PIN3* regulation | Main text figures | Supplementary Modelling Information | Supplementary Figures | Model is validated by experimental observations |
|---|---|---|---|---|---|---|
| 1 | Direct target of ARF7 and IAA14 | AEC/PIN3 not included | Figures 4G, E and 5C | Supplementary Modelling Figures M2–M13 | Supplementary Figures S2 and S4 | No |
| 2 | Direct target of ARF7 and IAA14 | Direct target of ARF7 and IAA14 | Figures 5D, E and 6A–C | Supplementary Modelling Figures M14–M25 | Supplementary Figures S5 and S6 | No |
| 3 | Indirect target of ARF7 and IAA14 (secondary response gene) | Direct target of ARF7 and IAA14 | Figure 6A | Supplementary Modelling Figures M14–M25 | Supplementary Figures S5 and S6 | Yes |

Note that model versions two and three have the same steady states and so some figures are valid for both models. IAA14: Aux/IAA protein acting as a negative regulator of ARF7; ARF7: Auxin response factor 7, a transcription factor that activates downstream auxin responsive genes; LAX3: auxin influx transporter; AEC/PIN3: Auxin Efflux Carrier PIN3.

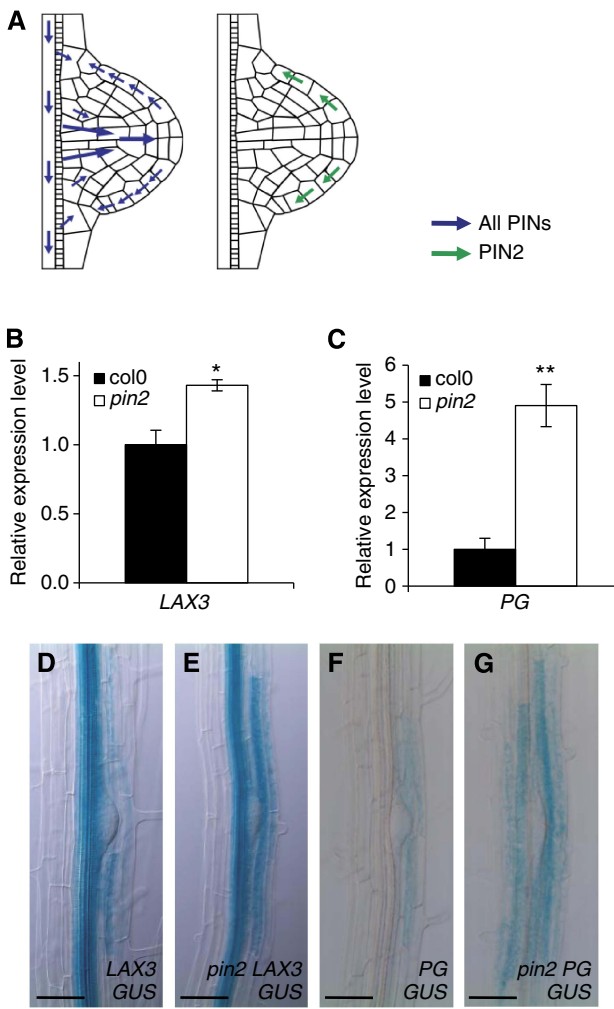

**All PINs**
**PIN2**

**Figure 2** The primordium acts as a source of auxin to induce genes involved in LR emergence. (**A–G**) The PIN2 auxin efflux carrier removes auxin from the apex of the LR primordium (**A**). As a result of the overaccumulation of auxin in the apex of the LRP in the *pin2* mutant background, both *LAX3* (**B**) and *PG* (**C**) are induced. Overaccumulation of auxin in the LRP tip in the *pin2* mutant background increases the strength but does not alter the expression pattern of *pLAX3:GUS* (**D**, **E**) and *pPG:GUS* (**F**, **G**). Asterisks indicate a significant difference with corresponding control experiment by Student's *t*-test (\**P*<0.05; \*\**P*<0.01). Bars are 50 μm (**D–G**).

and H). *J0121 ≫ iaaH-RFP* was crossed with the *pLAX3:LAX3-YFP* line and F1 progeny treated with or without IAM. In the absence of IAM, LAX3-YFP remained expressed in just two cell files (Figure 3J). In contrast, LAX3-YFP was induced in several cortical cells when co-expressed with *J0121 ≫ iaaH-RFP* and treated with different concentrations of IAM (Figure 3K and L). On the basis of our experimental results, we conclude that the XPP cells are able to channel auxin to induce *LAX3* expression in cortical cell files overlaying stage I primordia.

## Probing the regulation of LAX3 spatial expression using a 3D mathematical model

To understand how the initial *LAX3* expression pattern might form (up to stage I of LRP development), we developed a

mathematical model based on the simplest possible assumptions consistent with the available experimental data, the intention being to use the model to generate clear experimental predictions that we could then test. These assumptions include that *LAX3* expression is induced by auxin in the cortex but not in the endodermis or pericycle (Swarup *et al*, 2008); that auxin induction of *LAX3* leads to an increase in LAX3 protein; and that this then causes auxin influx activity to increase in *LAX3*-expressing cells. Since the wild-type expression pattern of *LAX3* is three-dimensional in nature (see Figure 1C–G and Supplementary Figure S1 for an example), the model incorporates cell-to-cell auxin transport in realistic 3D root cell and tissue geometries appropriate to stages 0–I of LRP development (Figure 4A–F; Supplementary Figure S1).

We began by developing an ordinary differential equation (ODE)-based model of the known regulatory network controlling *LAX3* induction (Figure 4A). The model comprises variables that represent the level of *LAX3* and *IAA14* mRNA and LAX3, IAA14 and ARF7 protein in each cortical cell. We initially assume that each gene in the network (namely *IAA14* and *LAX3*) is a direct target of ARF7 and IAA14. The rate of transcription of target genes is assumed to be an increasing function of ARF7 and a decreasing one of IAA14, so that IAA14 antagonises ARF7-mediated activation. The level of LAX3 and IAA14 protein is determined both by the rate of translation of their respective mRNAs and by degradation; the IAA14 degradation rate is modelled as an increasing function of the auxin level (see Middleton *et al*, 2010).

To capture the cell-to-cell movement of auxin, we adapted the auxin transport model first proposed by Kramer and co-workers (Swarup *et al*, 2005) so that it was suitable for three-dimensional vertex-based geometries (see also Grieneisen *et al*, 2007 and Laskowski *et al*, 2008 for further examples of models of auxin transport). The model takes into account pH differences between the apoplast (the extracellular space that joins two cells) and the cytoplasm, which result in acid trapping (whereby auxin can pass through cell membranes by diffusion, but require specific efflux transporters to leave the cell at a similar rate; see Supplementary Modelling Information for more details). We note that the model presented in Swarup *et al* (2005) also included intracellular and apoplastic auxin gradients. However, as noted more recently in Kramer (2006, 2008), diffusion of auxin in the apoplast is slow enough that its movement is dominated by carrier-mediated transport (see Kramer *et al*, 2007 for estimates of the diffusion coefficient in the apoplast). On the basis of the calculation presented in Kramer (2006), we estimated (using the default parameter values provided in Supplementary Modelling Information Table M2), that in the cortex or endodermis, an auxin molecule will travel up to ∼5 μm if a cell is not expressing the influx carrier, and less than a micron if it is expressing one. Importantly, these distances are independent of the rate at which the auxin enters the apoplast (see Kramer, 2006), and are relatively small when compared with the diameter of a cortical or an endodermal cell (∼10–15 μm). Interestingly, the apoplastic auxin diffusion rate is even slower in the epidermis than in the cortex or endodermis (by an order of magnitude; see Kramer *et al*, 2007). Similarly, as discussed in Kramer (2008), current estimates for the rate of auxin diffusion inside a cell indicate

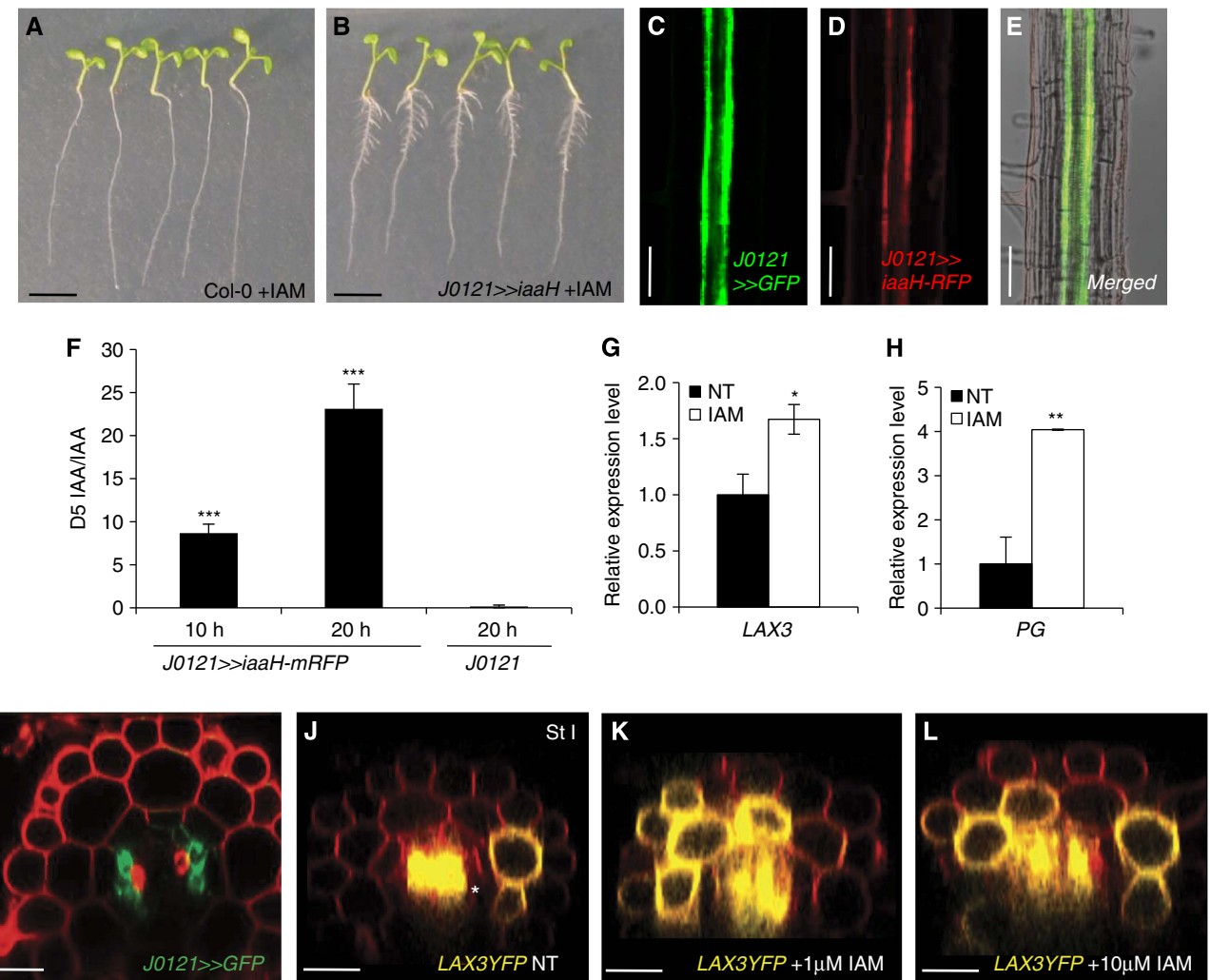

**Figure 3** Auxin moves from the pericycle cells towards the outer tissues. (A–L) Tissue-specific expression of the Agrobacterium indole-3-acetamide hydrolase (iaaH) gene was used as a tool to control auxin synthesis. Expression of the *iaaH* gene under the control of the J0121 GAL4 driver line (**B**) resulted in production of numerous lateral roots upon 1 μM indole-3-acetamide treatment whereas wild-type plants (Col-0) were not affected (**A**). Transgenic *Arabidopsis* plants were engineered to synthesise IAA in xylem-pole pericycle (XPP) cells using the XPP-specific GAL4 drive line, J0121 (**C**) by targeting the expression of the bacterial iaaH enzyme fused to the RFP tag (**D**). Therefore, the J0121 and iaaH fluorescent markers (respectively GFP and RFP) are co-expressed in the XPP tissue (**E**). Plant roots were fed with 5 μM deuterated indole-3-acetamide (D5-IAM), and the ratio of D5-IAA to IAA was determined by mass spectrometry analysis. A strong D5-IAA signal was detected upon 10 and 20 h after incubation in the J0121 $\gg$ iaaH-RFP line, whereas very little D5-IAA was detected in the control (J0121) line even after 20 h of incubation (**F**). As a result of auxin synthesis in the xylem-pole pericycle cells, we observed an overexpression of LAX3 (**G**) and PG (**H**). The J0121 line is specifically expressed in three pericycle cell files associated with each of the xylem poles as seen with the GFP reporter (**I**) and as reported previously (Laplaze *et al*, 2005). (**J–L**) We monitored the effect of auxin synthesis in the pericycle on the LAX3 expression pattern by crossing the J0121 $\gg$ iaaH-RFP plants with the *pLAX3:LAX3YFP*-expressing plants and subsequent analysis on the F1 progeny. LAX3-YFP accumulation was visualised upon 18 h treatment with mock (**J**), 1 μM IAM (**K**) or 10μM IAM (**L**). Note that strong expression of LAX3 in the vasculature is seen as in the non-treated control. Bars are 25 μm (**I–L**), 50 μm (**C–E**) and 5 mm (**A, B**). Data shown are mean value ± s.e.m. and n = 100 plants. Asterisks indicate a significant difference with corresponding control experiment by Student's *t*-test (*$P < 0.05$; **$P < 0.01$; ***$P < 0.001$).

that the intracellular gradient can form in long cells (of ~100 μm or more), but this refers to the case where efflux carriers are polarised at only one end of the cell, thus acting as a localised auxin sink. This occurs in the meristem and elongation zones of the root (the latter of which was the focus of the model presented in Swarup *et al*, 2005), where members of PIN family of auxin efflux transporters are polarly localised in shootward and rootward directions in specific cell files (see also Blilou *et al*, 2005). However, these expression patterns are not observed in the mature region of the root, where the LRP

start to emerge. Furthermore, LAX3 is expressed in an apolar manner in cortical cells (Figure 1). Therefore, according to these considerations, the concentration of auxin in both inside apoplastic and cellular compartments can be regarded as spatially uniform. Auxin fluxes due to diffusion and carrier (including LAX3)-mediated transport are therefore described using systems of ODEs, from which the intracellular and apoplastic concentrations of auxin can be calculated.

The root tissue geometries were generated using the following meshing pipeline. First, cross-sectional images

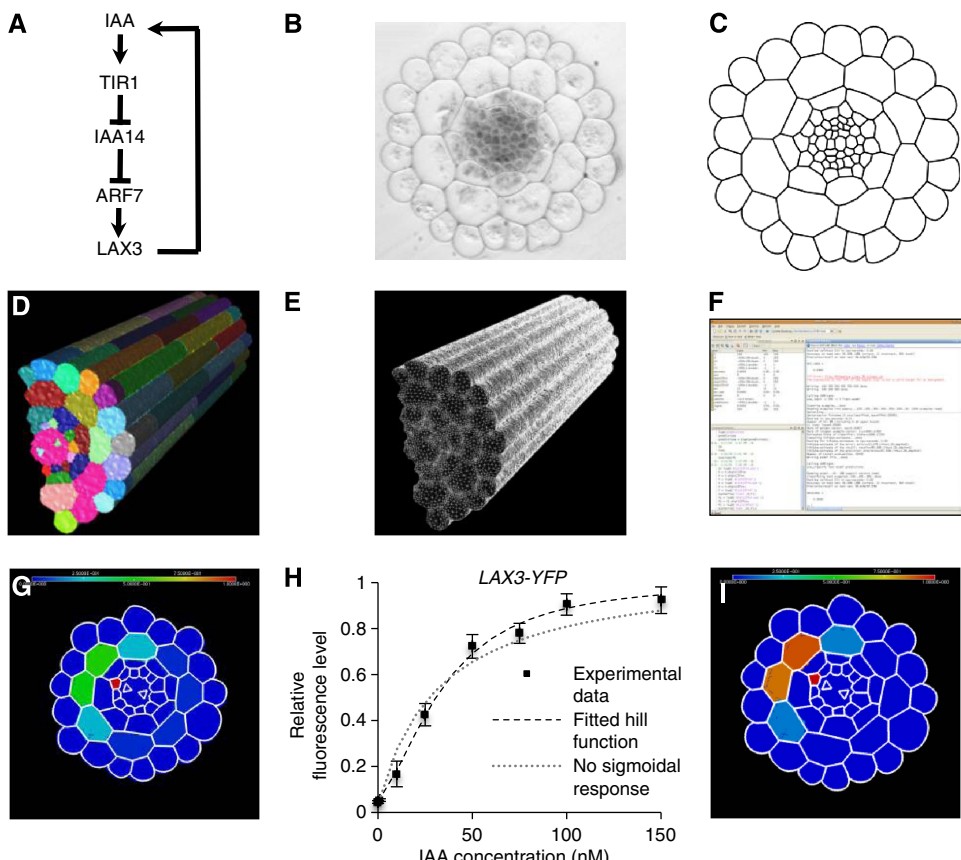

**Figure 4** Computational 3D model of *LAX3* induction by auxin in the cortical cells of the *Arabidopsis* root. (**A**) Representation of the gene regulatory network controlling lateral root emergence. IAA: auxin; TIR1: F-box protein that interacts with IAA14 to form the auxin receptor complex; IAA14: Aux/IAA protein acting as a negative regulator of ARF7; ARF7: Auxin response factor 7, a transcription factor that activates downstream auxin responsive genes; LAX3: auxin influx transporter. (**B–F**) The meshing pipeline. A cross-section of a resin embedded wild-type (Col-0) *Arabidopsis* root (**B**) was manually digitised using Adobe Illustrator (**C**) and then extruded to realistically recreate the third dimension (including transverse walls, see main text) (**D**). A triangular mesh of the 3D object surface was generated using the CGAL library (**E**) and the surface triangles were imported into MATLAB to run the simulations (**F**). (**G**) Model output showing diffuse LAX3 protein accumulation in four cortical cell files (as indicated by a heatmap scale ranging from 0 (blue) to 1 (red); simulations shown are for the case in which the auxin source is provided by cells in a single XPP file; indicated in red). (**H**) LAX3 auxin dose response was determined by confocal microscopy of the LAX3-YFP protein and fitted to a Hill function (best fit: Hill Coefficient = 2; threshold = 40 nM); if no sigmoidal response was assumed (in the sense that the Hill Coefficient = 1), then the fitted curve consistently overestimated the LAX3 response at low levels of auxin and underestimated it at high levels of auxin. Values are plotted as mean ( ± s.e., *n* = 3). (**I**) Including the sigmoidal response increased the sharpness of the predicted *LAX3* expression pattern, but still shows expression in four cortical cells. Scale bar is 25 µm (**B**).

of *Arabidopsis* roots were captured using light microscopy (Figure 4B). These were then manually segmented (Figure 4C) and then extruded to recreate the longitudinal axis artificially (Figure 4D; Supplementary Figure S1). This included adding cell walls in the longitudinal direction to match experimentally observed cell lengths (epidermis: 200 µm, cortex: 160 µm, endodermis: 40 µm and pericycle: 80 µm) and organisation (so that cells in different files are arranged in a staggered formation, illustrated in Figure 1B and Supplementary Figure S1). Triangular meshes were generated using the CGAL library (Figure 4E). The mesh was then imported into MATLAB (Figure 4F), which was used to compute key parameters of the auxin transport model (cell volumes, cell type, cell–cell surface areas and cell–neighbour relationships).

The coupling between our auxin transport model and the regulatory network (summarised in Figure 4A) appears through LAX3 affecting auxin transport and hence intercellular accumulation. This in turn affects the abundance of Aux/IAA repressor proteins in the gene network submodel, which then

impacts on *LAX3* mRNA (and hence protein) levels. Solutions to the model were computed in MATLAB. Finally, the computer package MEDIT (Frey, 2001) was used to visualise the model outputs (Figure 4G).

To ensure that our results are robust to natural variations in root tissue geometry, all simulations were repeated using three-dimensional meshes generated from cross-sections of three different roots. However, each root contains two sets of XPP cell files, either of which could develop into LRP (see Figure 1A). We therefore ran simulations where cells from either one of the two XPP files act as a source. This effectively doubled the total number of different geometrical arrangements tested to a total of six. All of the carriers included in the model, and their associated polarities, are illustrated in Supplementary Modelling Figure M1. In particular, since the manner in which auxin move from XPP source cells to the cortex is currently unknown, we assumed that auxin is directly transported from the pericycle to the endodermis, and then to the cortex. We discuss how our model results depend on these assumptions in Conclusion.

Where appropriate, we illustrate results using three-dimensional images of our simulations, where we have artificially removed the epidermis so the cortex is visible (Supplementary Figure S1). However, some figures use two-dimensional cross-sections of the 3D simulations for visual clarity (with the position of the cross-section in the model is highlighted in Supplementary Figure S1, this being chosen so that it is consistent with the position of the cross-section used in Figure 1A–D). For each geometry and model variant, we determined which cells are predicted to express LAX3, and from these calculate frequencies analogous to those presented in Figure 1 (see below for further details).

As noted above, the model is intended to capture LAX3 expression during stages 0–I of LR development. During these stages, XPP cells (from which LRP originate) undergo several rounds of anticlinal division (see Figure 1B). Here, auxin responses can be detected in all the daughter cells of the same XPP file (Benková *et al*, 2003). To see whether these divisions impact on the behaviour of the model, we compared simulations where these divisions took place to the case where they did not, and found that the respective model outputs were barely distinguishable (see Supplementary Modelling Figure M2). This is because the total volume of the daughter cells is always contained within the total volume of the parents, and so from the perspective of the overlaying tissues the nature of the auxin source does not significantly change. We therefore do not include these divisions in the model for the simulations discussed here.

## Regulation of *LAX3* by auxin alone cannot explain *LAX3* spatial expression

On the basis of our experimental results (Figures 2 and 3), we first assumed that the XPP cells primed to form LRP provide a source of auxin, and that this emanates from cells in either all three of the XPP files or only the middle XPP file (see Figure 1A and B). We used the model (namely model version one in Table I) to test both scenarios. To compare the model outputs with the experimental data, we first calculated how the steady-state levels of LAX3 expression in individual cells vary according to the magnitude of the auxin source (see Supplementary Modelling Figure M7). We assumed that the auxin supply rate of each LRP is approximately the same for each root, so that on average LAX3 expression is at least 50% of the maximum possible level. We consider a cell in the model to express LAX3 if its abundance levels are above 5% the maximum (i.e., so LAX3 in non-expressing cells is at most one tenth of the level found in the expressing ones). We then used these criteria to generate frequencies for the number of cells expressing *LAX3*, analogous to those presented in Figure 1.

Our simulations revealed that, regardless of whether auxin originated from cells in just one or all three XPP files, the model could not capture the spatially restricted *pLAX3:LAX3-YFP* expression pattern. Instead, for each of the tissue geometries tested, when auxin originates just from one XPP file, *LAX3* would be expressed in typically three or more cortical cell files (83%; Supplementary Figure S2A), although in some cases only two cell files would express LAX3 (16%; Supplementary Figure S2A). None of the geometries tested predicted that LAX3 expression was restricted to just one cell

file (as is observed experimentally; Figure 1E). In the case where the source emanates from three XPP files, the cortical *LAX3* expression pattern was predicted to typically span three or more cell files (100%; Supplementary Figure S2B). In the longitudinal direction, expression was predicted to occur in one cell per file for 50% of the geometries tested or two cells per file in the remaining cases. This was broadly similar with the observed frequencies of 58.9% for one cell per file or 41.1% for two cells per file (see Figure 1F). Thus, the strongest discrepancy between the data and the model occurred in the circumferential direction. Closer inspection of the different tissue geometries used revealed that there is a strong root-to-root variation in the number of cortical cell files to which the source cell (or cells) makes indirect contact via the endodermis (see Figure 1A). Thus, depending on the tissue geometry, a single XPP file makes indirect contact with two (16%), three (50%) or four (33%) cortical cell files (Supplementary Figure S2C); three XPP cell files make indirect contact with either four (50%) or five (50%) cortical cell files (Supplementary Figure S2D). In general, cells that make this indirect contact are predicted to express LAX3.

Our results suggested that additional regulatory mechanisms are likely to be important if the *LAX3* spatial expression pattern is to be robust to natural variations in tissue geometry. We therefore explored whether *LAX3* spatial expression relies on the relationship between auxin concentration and its induction. Quantification of LAX3-YFP induction at different auxin concentrations revealed a sigmoidal response. However, it only had a Hill coefficient of ∼2 (Figure 4H; Supplementary Modelling Information) and was only moderately switch-like. The Hill coefficient measured here provides quantitative information about how the LAX3 regulatory network responds to exogenous auxin, but does not directly correspond to any one biochemical reaction in the network. For this reason, we refer to it as the 'effective' Hill coefficient. However, individual parameters in the regulatory network model can be chosen to capture the observed response of LAX3 to exogenous auxin (see Supplementary Modelling Figure M3). When extended to do so, the model was still unable to capture robustly the wild-type *LAX3* spatial expression pattern. Instead, we observed that, for a single auxin source file, *LAX3* was still typically expressed in three cell files (66%; Supplementary Figure S2E; Supplementary Modelling Figures M8–M10), although its expression pattern was notably sharper (Figure 4I) than before (see Figure 4G). Similarly, in simulations involving three XPP source cells, we observed that *LAX3* was expressed in three or more of the cortical cell files for the geometries tested (Supplementary Figure S2F; Supplementary Modelling Figures M11–M13). For none of the geometries tested *LAX3* expression was restricted to just one cell file.

Although our simulations of the model indicated that it is unable to account for the stereotypical *LAX3* expression pattern, this could be due in part to the parameter values used. Fortunately, estimates are already available for parameters associated with auxin transport (see Supplementary Modelling Information for further details). However, the LAX3 regulatory network is not yet parameterised. To address this issue, we reduced the complexity of the model by noting that movement of auxin is dominated by

carrier-mediated transport. Our reduced version of the model, although considerably simpler, could accurately capture the behaviour of the full model (see Supplementary Modelling Figures M4 and M5) while now being amenable to mathematical analysis (see Supplementary Modelling Information for details). In particular, we noted that since auxin is effectively trapped inside the cortical cells once it enters them, there is only negligible communication between neighbouring cortical cells. Moreover, due to lack of carrier activity transporting auxin from cortical cells to endodermal ones, movement of auxin between endodermal and cortical cells is essentially one way. Thus, accumulation of auxin in the endodermis can be thought of as an 'input pattern', and the LAX3 expression pattern in the cortex can be regarded as the 'output'. In particular, the level of auxin received from the connecting endodermal cells by the overlaying cortical cells depends in large part on the area shared between them. The arrangement between these cells can be rather asymmetric, so that there is one long edge connecting a cortical cell file to an endodermal one, together with one or two shorter edges connecting the other cortical cell files (where the long edge is, on average, approximately three times longer than the short one; see Supplementary Figure S3). Thus, of the cortical cell files that connect (indirectly) to a XPP source cell, some will do so only via a short edge and others via a long edge. We refer to the former as 'minor' cortical cell files and the latter as 'major' cortical cell files.

In all the tissue geometries tested, the central source cell connects to two endodermal cell files, and from these to two major cortical cell files (and several minor ones; see Figure 1A). If the auxin source emanates from all three XPP files, then the number of major cortical cell files varies from three to four. Thus, we reasoned that if this model version is to account for the fact that *LAX3* expression is limited to at most two cell files (Figure 1E), it must then satisfy the following conditions. First, auxin only emanates from the central source file, and the supply rate is high enough for *LAX3* expression levels to attain 50% maximum in a major cortical file. Second, *LAX3* is not expressed in the neighbouring minor cell files (i.e., below 5% of the maximum). From this, we then calculated how sharp the LAX3 response to auxin must be (i.e., how large the effective Hill coefficient should be) to satisfy these two conditions, using the reduced model equations (see Supplementary Modelling Information). According to our analysis, the shape of the LAX3 response function can be measured directly from exogenous auxin treatments (as are provided in Figure 4H, see also Supplementary Modelling Figure M5). Accordingly, we found that even if *LAX3* expression levels in the minor cortical cell files were allowed to be at 10% of their maximum and not 5% (and so the level of LAX3 in non-expressing cells is one fifth that in the expressing ones), the effective Hill coefficient would need to be set to at least 4 (twice that was measured, see Figure 4H). If *LAX3* expression in the major cortical cell file was higher than 50% (so that the differential *LAX3* expression between neighbouring cortical cells was even greater), then we would require an even larger effective Hill coefficient to compensate. Thus, for the current version of the model to account for the stereotypical *LAX3* expression pattern, the LAX3 response to exogenous auxin has to be considerably sharper than that observed experimentally (see Figure 4H).

## Multicellular models reveal a role for the auxin efflux protein PIN3 in *LAX3* patterning

The difference between our experimental observations and model predictions motivated us to reassess our model assumptions. One possibility was that we were missing additional auxin-transport components. To test this possibility, we treated LAX3-YFP roots with the auxin influx carrier inhibitors 1-NOA or 2-NOA (Figure 5A and B; Supplementary Figure S4) and compared them with our simulations where LAX3-mediated transport is blocked *in silico* (Figure 5C–E). Using the model, we predicted that *LAX3* expression would be restricted to just a few cells. However, our experimental results contradicted the model prediction and showed that after 1-NOA or 2-NOA treatment, cortical LAX3-YFP expression gradually spread circumferentially and longitudinally from just the original one-two cell files in which it was being expressed to all cortical cells (Figure 5A and B; Supplementary Figure S4B). Disruption of polar auxin transport is known to alter auxin levels in whole plants (Ljung *et al*, 2001), and the magnitude of the spread indicated that the level of auxin being supplied by the primordia might have also been affected (indirectly) by the NOA treatment. However, even if auxin supply rates were increased 20-fold, *LAX3* expression was restricted to three cortical cell files (83%) when there was just one source file, or four or more (100%) if there were just three source files (Supplementary Figure S4A; Figure 5C). In general, *LAX3* expression was restricted to only those cortical cells making indirect contact (via an endodermal cell) with an XPP source cell (see Figure 1A): only for extremely large increases in the auxin supply rate (200-fold or more) would *LAX3* be expressed in additional cortical cells (albeit at a very low level). As noted previously, diffusion of auxin in the apoplastic region connecting the cortical and endodermal cells may allow the auxin to travel slightly further than predicted here (approximately a third of a cortical cell diameter, see above). Since this estimate is independent of the rate at which auxin enters the apoplast, and hence independent of magnitude of the auxin source, we concluded that our model must lack a key component that facilitates auxin transport to other cortical cells.

We next considered the possible role of auxin efflux carriers, and modified the model to investigate how the presence of an auxin inducible efflux carrier would affect the *LAX3* expression pattern (see Supplementary Modelling Information). Since the direction of the LAX3 spread occurred in both circumferential and longitudinal directions, we assumed that the efflux carrier was polarised towards neighbouring cortical cells. We then ran simulations in which LAX3 activity is inhibited (but assuming that the auxin supply rate by the primordia was unaffected) and the auxin efflux carrier is induced by auxin. Unlike before, LAX3 expression would spread to tens of cells both longitudinally and circumferentially, so that it was now expressed in cortical cells not making contact (via a single endodermal cell) with an XPP source cell. However, the range of the spread appeared to be larger in our experiments than was predicted by our model, which we assume is due to the NOA treatment indirectly causing an increase in the rate of auxin supply emanating from the primordia. Unlike the case when no efflux carrier is included in

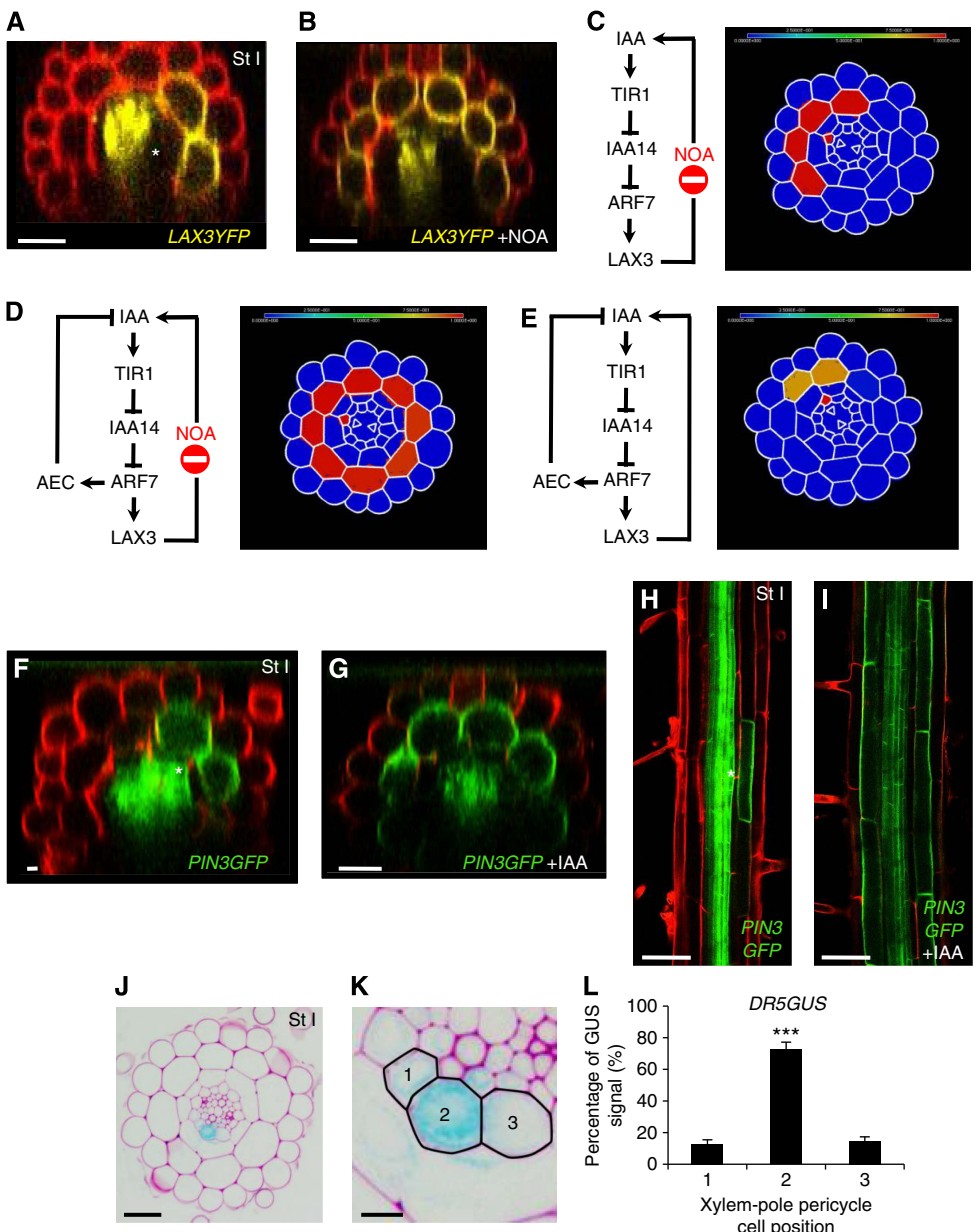

**Figure 5** Computational 3D modelling of *LAX3* expression predicts the involvement of the auxin efflux transporter PIN3. (**A–D**) Accumulation of LAX3-YFP in the non-treated conditions (**A**) or after 18 h treatment with 10 μM 2-NOA (**B**). Simulation of LAX3 induction in the presence of auxin influx inhibitor NOA in the absence of an auxin-induced efflux carrier (AEC) shows expression of LAX3 to be limited to a few cell files, in contrast to (**B**). In the presence of both an auxin influx inhibitor NOA and an AEC, *LAX3* expression is predicted to spread throughout the cortex. When an AEC is included, the model can capture the stereotypical wild-type pattern (**E**). (**F–I**) The auxin efflux carrier PIN3 is expressed in the cortical cells situated in front of the LRP and is induced by auxin in this tissue. Accumulation of PIN3GFP in the non-treated condition (**F**, **H**) and upon 18 h 1 μM IAA treatment (**G**, **I**). Asterisks indicate the presence of LRP in the non-treated conditions. IAA: auxin; TIR1: F-box protein that interacts with IAA14 to form the auxin receptor complex; IAA14: Aux/IAA protein acting as a negative regulator of ARF7; ARF7: Auxin response factor 7, a transcription factor that activates downstream auxin responsive genes; LAX3: auxin influx transporter. (**J–L**) Cross-section of a DR5:GUS stained root showing a stage I primordium at low (**J**) and high magnification (**K**). Quantification of the DR5GUS signal (blue channel) in the three stained xylem-pole pericycle cells (numbered from 1 to 3 as shown in K) using ImageJ (**L**). Data shown are mean ± s.e.m. and $n = 10$ sections. Asterisks indicate a significant difference with corresponding control experiment by Student's *t*-test (*** $P < 0.001$; $n = 10$). Scale bars are 50 μm (**H**, **I**), 25 μm (**F**, **G**), 20 μm (**J**) and 5 μm (**K**).

the model, a 20-fold increase in the auxin supply rate, combined with a blocking of LAX3 function, caused all cells to express *LAX3*, as was observed experimentally (Figure 5D). This was the case regardless of whether the auxin source is from one pericycle cell or three.

In light of these results, we tested whether there was any experimental evidence supporting the presence of an auxin

efflux carrier in the cortical cells in front of LRP. Confocal imaging of GFP reporters fused in frame to coding sequences of either PIN or ABCB classes of auxin efflux carriers revealed that *PIN3* exhibited such a pattern of expression (Figure 5F and H). Our model also predicted that the auxin efflux carrier is auxin inducible in cortical cells. Auxin treatment confirmed that this was indeed the case for *PIN3* (Figure 5G and I).

Interestingly, the PIN3-GFP protein appeared to localise preferentially in the lateral, distal, shootward and rootward faces of the cortical cells, whereas less accumulation is seen in the proximal face of the cortical cells (Figure 5F–I), in accordance with its potential role in moving auxin towards the outer tissues.

## The central XPP cell is the dominant source of auxin in LRP

We next ran simulations of the new model (this being model version two, in which PIN3 and LAX3 together regulate the transport of auxin) and analysed how the steady-state *LAX3* expression pattern depended on the magnitude and spatial distribution of the auxin source. Interestingly, we found that as the magnitude of the auxin source was increased, *LAX3* expression levels would now vary in a switch-like manner (reminiscent of having a very high Hill coefficient, as was required by our analysis of model version one; see Supplementary Modelling Figures M14 – M16 for representative examples), even though the predicted shape of the auxin dose response experiments was unaffected by the inclusion of PIN3 (i.e., and still produced the same sigmoidal shape with effective Hill coefficient of two, as in model version one and Figure 4H). We understand this as follows. Expression of cortical PIN3 means that (in contrast to model version one) auxin is able to move from cortical cell to cortical cell, ensuring that there is strong cell–cell communication. Interplay between LAX3 and PIN3 can thus create sharp local intercellular gradients of LAX3 expression when induced by a localised auxin supply (i.e., by the LRP). However, when performing whole tissue auxin dose response experiments, these localised gradients are effectively eliminated, along with the switch-like induction properties of LAX3 (see Supplementary Modelling Figure M3). Thus, the model has helped us resolve the apparent discrepancy between the rather weak switch-like response of LAX3 to exogenous auxin (Figure 4H) and its strikingly discrete expression pattern (Figure 1).

For the case where the auxin source is provided by just one source XPP file, the model predicts that *LAX3* expression was restricted to just two cell files, provided the magnitude of the LAX3 source was chosen within a certain range (with this range varying from root to root; Supplementary Modelling Figures M14–M16). For some of the tissue geometries tested, the model is also bistable (Supplementary Modelling Figures M14–M16), such that (for certain ranges of the auxin supply rate) either two or three of the cell files could be selected to express LAX3 (see later for details). By counting the number of cells that express *LAX3* according to the criteria we established for model version one (see Materials and methods), *LAX3* would be strongly expressed in just one cell file for 33% of the geometries tested (and in two cell files otherwise; see Supplementary Modelling Figures M14 and M15 and Supplementary Figure S5A). This compares well with the corresponding occurrence rate of 32.6% observed *in planta* (Figure 1E). Thus, by including PIN3 in the model we can also explain why LAX3 is sometimes only expressed in one cell file; this was not the case with any of the previous model formulations. As with the previous model version,

the number of cells per file ranged from one to two, as was observed experimentally (Figure 1G; Supplementary Modelling Figures M14–M16). In summary, the new model can account for the stereotypical wild-type *LAX3* expression pattern (Figures 1 and 5A), as long as the auxin source is provided by just one XPP cell file. However, when all three XPP files provide the auxin source, LAX3 was typically expressed in three cortical cell files (88%), an occurrence which is rare *in planta* (2.3%) (Supplementary Figure S5B; Supplementary Modelling Figures M17–M19). This should not be surprising, since in this case the XPP source cells make indirect contact with typically three major cortical cell files.

On the basis of our model predictions, we tested whether cells in central XPP file were the dominant source of auxin, by monitoring the spatial expression of the auxin response marker *DR5:GUS* reporter line in radial cross-sections of new LRP. Upon LR initiation, auxin response occurs in cells in all three XPP files (Figure 5J and K), Nevertheless, the central cell file showed a much stronger *DR5:GUS* signal than the two lateral cell files. Quantification of the GUS signal demonstrated that 73% was observed in the central cell file compared with 12–15% for each side file (Figure 5L). Hence, the model has helped reveal the presence of a new network component (PIN3) and the importance of a spatially restricted auxin source (Figure 6A).

## PIN3 and LAX3 are expressed consecutively during LR emergence

We next analysed the dynamics of the model further. We found that if LAX3 is induced at approximately the same rate as PIN3, LAX3 would spread circumferentially before ultimately being restricted to a few cell files (as described above). However, this transient behaviour was not observed if LAX3 was induced at a much slower rate than PIN3 (Figure 6B and C). In addition, in cases where the model was bistable (so that either two or three cell files could be selected—see above), we found that the steady state to which the system tends depends again on the relative timing of LAX3 induction. If *LAX3* is induced at the same rate as *PIN3*, then three cell files are selected for expression; if *LAX3* was induced at a slower rate than *PIN3*, then two cell files are selected (see Figure 6B and C respectively). Together, our observations suggest that *PIN3* and *LAX3* need to be induced consecutively.

To test this model prediction, we conducted a detailed time course of *LAX3* and *PIN3* response to auxin treatment (Figure 6D and E) to observe whether *LAX3* and *PIN3* mRNAs are induced at different rates. Consistent with our predictions, we found that auxin induction of *LAX3* expression was slower when compared with *PIN3* (Figure 6D and E). We therefore hypothesised that *LAX3* is a secondary auxin responsive gene. To test this possibility, auxin time-course treatments were performed in the presence or absence of the protein synthesis inhibitor, cycloheximide (CHX). In the presence of CHX, *PIN3* remained auxin inducible (Figure 6D) demonstrating that it is a primary auxin responsive gene. In contrast, in the presence of CHX, *LAX3* expression was not induced by auxin, consistent with it being the secondary auxin responsive gene (Figure 6E), as anticipated by our modelling.

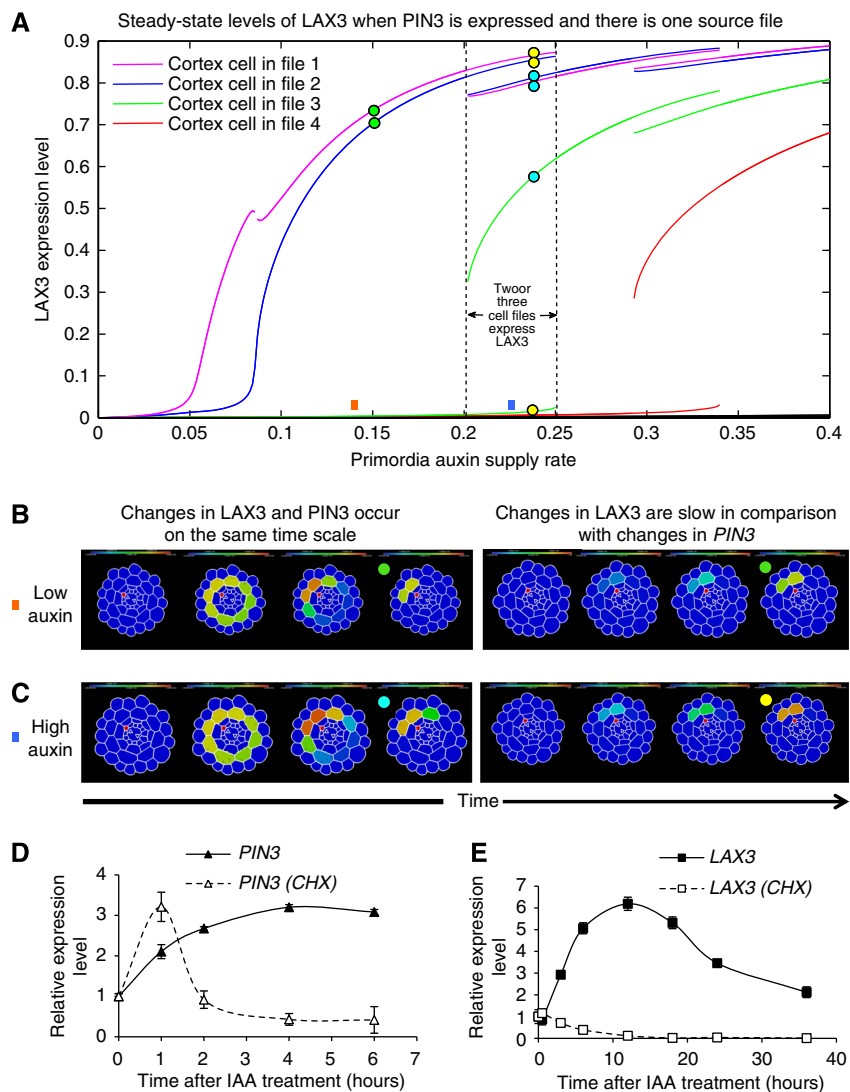

**Figure 6** *PIN3* and *LAX3* are sequentially induced to promote lateral root emergence. (**A**) Steady-state solutions to model version two (wherein both LAX3 and PIN3 are expressed, see Figure 5E and Table I) for various LRP auxin supply rates. For a supply rate within a certain range, LAX3 expression is restricted to just two cell files (as indicated). Within this range, the levels of LAX3 in other cell files remain low (compare with analogous plots obtained for model version one shown in Supplementary Modelling Figures M8–M10). For slightly higher supply rates, LAX3 can be expressed in three cell files at steady state. Overlap between the two regions corresponds to bistability. (**B**, **C**) Visualisation of model dynamics for various auxin supply rates (as indicated by square boxes) and *LAX3* induction rates. Together, these determine the steady state to which the model tends (as indicated by the coloured circles). (**B**) If the auxin supply rate is chosen so that the model has one stable-steady state, and *LAX3* and *PIN3* are induced by auxin at the same rate, then LAX3 expression initially spreads circumferentially before being restricted to two cell files. (**C**) If the auxin supply rate is chosen so that the model is bistable, then *LAX3* expression spreads throughout the cortex before being restricted to three cell files. In either case (bistable or monostable), if *LAX3* is induced slowly by auxin, then the transient spread is eliminated and only two cell files are selected for LAX3 expression. (**D**, **E**) PIN3 but not LAX3 is a primary auxin response gene. Induction of *PIN3* is observed from 1 h onward and is not blocked by treatment with the protein synthesis inhibitor cycloheximide (CHX) (**D**). Induction of *LAX3* is detected from 4 h onward and is totally blocked by CHX treatment (**E**). Data shown are mean value ± s.e.m. ($n = 100$ plants).

## Interplay between PIN3 and LAX3 sharpens the *LAX3* expression pattern

We updated our model (to model version three) by treating *LAX3* as the secondary response gene rather than the primary response. It is possible to map exactly the parameters of model version two onto model version three so that they have the same steady states (see Supplementary Modelling Information). Thus, treating *LAX3* as the secondary response gene alters only the dynamics of the system, namely by eliminating its transient spread throughout the cortex (as described above). In the model, interplay between the LAX3

and PIN3 feedback loops allows the selection of specific cell files for *LAX3* expression, and allows for a high level of robustness towards natural variations in tissue geometry. However, if we impair both LAX3 and PIN3 functionality in the model (thus removing both the feedback loops) we found that LAX3 expression would typically spread to three or more of the cell files (83 %; Supplementary Figure S6A and Supplementary Modelling Figures M20–M25). To test this, we performed experiments where both NOA and the auxin efflux inhibitor NPA were applied simultaneously and monitored changes in LAX3-YFP over time. Consistent with our predictions, *LAX3*

expression spread from two cell files to three cell files (Supplementary Figure S6C and D).

## Discussion

In recent years, research in the life sciences has focused on identifying the key molecular players (genes, proteins, RNAs, etc.) that regulate cellular behaviour. However, interactions between these components can form complex regulatory networks, the dynamics of which are difficult to predict through intuition alone (Middleton *et al*, 2012). In the case of multicellular organisms such as plants, the regulatory networks operate not only at molecular scale, but must also integrate information at the cell and tissue scales. The use of mathematical models, in conjunction with experimental approaches, will become increasingly important as our ability to generate data on the various pathways and regulatory mechanisms improves.

Here, we have illustrated how mathematical modelling, combined with traditional experimental approaches, can help us to unravel the mechanisms underlying complex developmental processes such as LR emergence. In particular, this study reveals how the sequential induction of auxin influx carrier and efflux carriers (Figure 7A) can ensure that only specific cells are targeted for cell separation. Here, auxin acts as an inductive signal that moves from the inner tissue of the root towards the outer tissue to trigger the LRP emergence response. As auxin enters cortical cells overlaying new LRP, this signal first activates *PIN3* expression (Figure 7B). This creates a flux of auxin towards the epidermis. While this leads to a net loss of auxin from the cortex, its level will be maintained in the major cortical cells (Supplementary Figure S3). For these, the intracellular auxin concentrations will be high enough to lead to subsequent induction of LAX3 (Figure 7C), leading to further accumulation of auxin into those cells. As a result, the *pin3* loss-of-function mutant is expected to show impaired LR emergence. Consistent with this prediction, we observed that the rate of LRP emergence was delayed in the *pin3-1* mutant background compared with control plants (Supplementary Figure S7).

These new mechanistic insights into the regulation of LR emergence have relied heavily on the use of mathematical models that integrate information about gene regulatory network, auxin transport and realistic tissue geometries. We had previously hypothesised that the localised source of auxin from newly initiated LRP and the positive feedback loop between *LAX3* induction and auxin accumulation would be sufficient to account for its striking spatial expression pattern (Swarup *et al*, 2008). However, when incorporating these details into a model with realistic tissue geometries, we found that this was not enough to explain the stereotypical LAX3 expression pattern. In particular, the number of cell files that expressed LAX3 varied strongly depending on which particular tissue geometry was used and on the magnitude of the LAX3 source (Supplementary Figure S2A and B). Mathematical analysis of the model revealed that, for to be consistent with the available data, the effective Hill coefficient for auxin induction of LAX3 would have to be much higher than that measured experimentally (Figure 4H). This is because, in

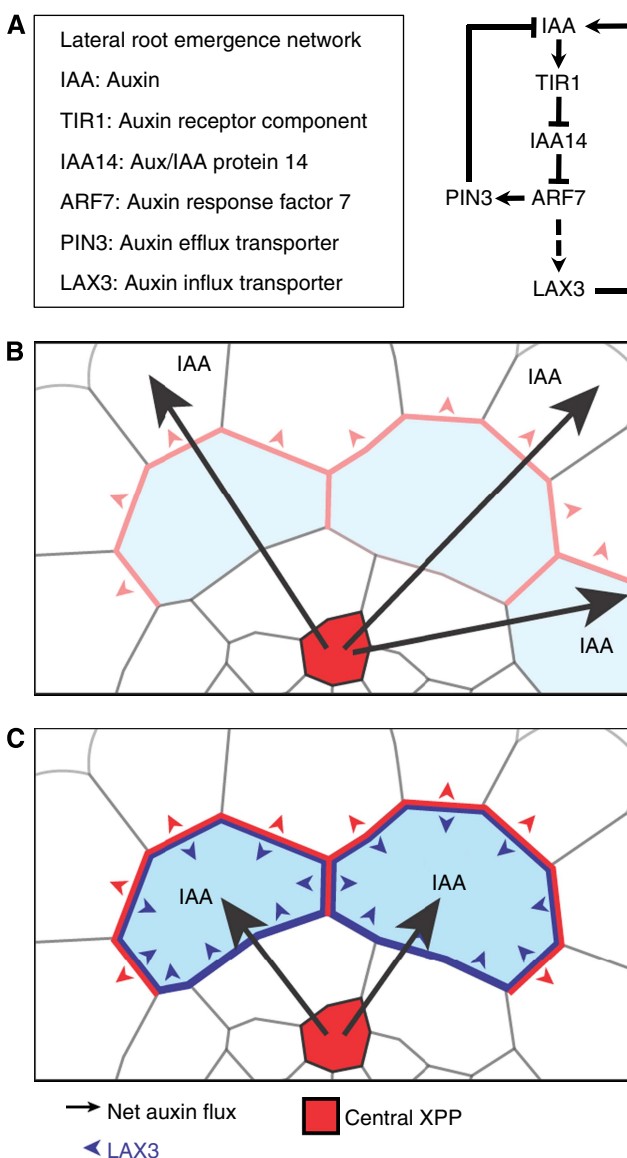

**Figure 7** Sequential action of PIN3 and LAX3 determines the specific expression pattern of LAX3 during lateral root emergence. (**A**) Regulatory network controlling LAX3 induction by auxin during lateral root emergence. (**B**) Auxin moves from the central xylem-pole pericycle file (XPP, red) towards the outer tissue and activates PIN3 in the cortex, resulting in a net flux of auxin towards the epidermis (red arrowheads). (**C**) The two files expressing PIN3 that make the most (indirect) contact with the cortex then accumulate enough auxin to induce *LAX3* at a later stage (blue arrowheads). Accumulation of LAX3 then leads to further increases in auxin levels, which subsequently trigger cell separation and promote the passage of the LRP.

model version one, auxin movement in the cortex is dominated by LAX3-mediated transport. Here, the resulting *LAX3* expression patterns are largely determined by how much auxin enters from the XPP source cells via endodermis (and not, for instance, by communication between cortical cells). Further discrepancies between the model outputs and our experimental observations led us to identify PIN3 as a key network component (Figure 5).

When PIN3 was included in the model (model version two), our steady-state analysis indicated that it could account for the stereotypical *LAX3* expression pattern (Figure 6), but only when auxin was being provided by just one XPP cell file. In particular, that *LAX3* is sometimes expressed in only one cell file. This is striking since none of the previous model formulations could account for this. Here, interplay between *PIN3* and *LAX3* can create sharp intercellular gradients in LAX3 expression (Figure 6). This is due in large part to the fact that, by expressing PIN3, cells can communicate effectively with their neighbours, thereby allowing them to coordinate which of the cells is to express *LAX3* and generate a sharp response to the input auxin pattern provided by the underlying endodermis.

Consistent with our prediction that auxin is being supplied by the central XPP cell file, we observed that auxin responses are strongest in this file (Figure 5J–L). We note that, since it is currently unclear how auxin moves through the endodermis, we had originally assumed that for auxin to reach the cortex from the XPP source cells it is transported radially from XPP cell to endodermal cell, and then from here to the cortex. Thus, based on these assumptions, auxin accumulation in the endodermis is largely restricted just the two cell files making direct contact with the central XPP source file (see Figure 1A). However, one further assumption that we could have made was that auxin could also move circumferentially or longitudinally, from endodermal to endodermal cell. This would increase the spatial extent of the auxin input pattern to include even more endodermal cell files. In the context of our model, this corresponds to the case where the auxin source was being provided by all three XPP cell files—which we found could not account for the stereotypical *LAX3* expression pattern. Therefore, based on our results it seems likely that, during LRP emergence, circumferential or longitudinal (but not radial) movement of auxin in the endodermis is negligible.

It has recently been found that expression of PIN3 in the endodermis has a role in LR development. Here, PIN3 in the endodermis is localised towards the pericycle, thus creating a reflux loop between the developing LRP and the overlaying tissues. Interestingly, we never observed endodermal PIN3 in our experiments (Figure 5F). This could be because the LRP in Marhavý *et al* (2013) (unlike here) are mechanically induced, and mechanical signals can lead to changes in PIN polarity and potentially in gene expression (Ditengou *et al*, 2008; Hamant *et al*, 2008). Thus, the mechanical induction of PIN3 in the endodermis could reflect a different development response to a distinct stimulus. Nevertheless, since the endodermal PIN3 in this case would act by transporting auxin back towards the primordia, in the context of our model this would simply reduce the magnitude, but not the spatial extent, of the auxin input pattern provided by the endodermis. For this reason, we do not expect inclusion of endodermal PIN3 in the model to significantly impact our results. Finally, we note that *LAX3* is expressed in the epidermis at much later stages of LR emergence than considered here (see Swarup *et al*, 2008), and it is likely that *PIN3* is also expressed in the epidermis during these stages as well.

Finally, studying the dynamics of our model proved critical and led us to discover that the efflux carrier *PIN3* and then the influx carrier *LAX3* must be induced sequentially. Otherwise, if *LAX3* was induced at the same time as *PIN3*, the former would be expressed (transiently) in all cell files (Figure 6B and C). Only when the induction of *LAX3* is slow compared with the induction of *PIN3* two cortical cell files will be selected (Figure 6B and C), consistent with the wild-type expression pattern (Figure 1). The functional importance of such temporal regulation during hormone responses has received very limited discussion in the plant field (Kuppusamy *et al*, 2009), but is evidently crucial to ensure a robust output from signalling pathways.

## Materials and methods

### Generation of plant materials

A multisite gateway approach (Karimi *et al*, 2007) was used to create the *UAS:iaaH-mRFP* line, allowing us to combine the three following ENTRY vectors pEN-UAS, pEN-iaaH and pEN-mRFP. pEN-UAS and pEN-mRFP were obtained from the VIB Gateway department. The *iaaH* gene was amplified from the *CAB:TMS2* line (Karlin-Neumann *et al*, 1991) by PCR (iaaHfor 5′-GGAGATAGAACCATGGTGGCCAT TACCTCGTT-3′ and iaaHrev 5′-GGGTCACCGCCTCCGGATCCATTGGG TAAACCGGCAAAAT-3′) and subsequently cloned into pDONR 221 to generate pEN-iaaH. Construct was verified by sequencing and used to transform *Arabidopsis* plants by floral dipping. Homozygous *J0121 ≫ iaaH-mRFP* lines were obtained after crossing with J0121 and selfing. Homozygous *J0121 ≫ iaaH-mRFP* plants were subsequently crossed with a *pLAX3:LAX3YFP* line and *LAX3* expression analyses were performed on F1 plants. Seeds for the following lines were obtained from the Nottingham *Arabidopsis* Stock Centre (NASC): J0121 (Laplaze *et al*, 2005), J0192 (Laplaze *et al*, 2005), *pLAX3:LAX3-YFP* (Swarup *et al*, 2008), *pin2* (Müller *et al*, 1998) and *pin3-1* (Friml *et al*, 2002). Plants were grown on vertical half MS plates at 23°C under continuous light (150 μmol m$^{-2}$ s$^{-1}$).

### Hormonal treatments

Plants were transferred on vertical half MS plates supplemented with IAA or IAM at a default concentration of 1 μM. Inhibition of protein synthesis was obtained by treating the plants with 50 μM cycloheximide (CHX).

### Quantitative RT–PCR

qRT–PCR was performed according to the rules previously published (Udvardi *et al*, 2008). Total RNA was extracted from 7-day-old roots ($n > 100$) using the RNeasy kit (Qiagen) with on-column DNAse treatment (RNAse free DNAse set, Qiagen). Poly(dT) cDNA was prepared from total RNA using the Transcriptor first-strand cDNA synthesis kit (Roche). Quantitative PCR was performed using SYBR Green Sensimix (Quantace) on a Roche LightCycler 480 apparatus. Target quantifications were performed with specific primer pairs (Supplementary Figure S8). Expression levels were normalised to the ubiquitin-associated gene UBA (At1g04850) using the following primers UBA forward 5′-AGTGGAGAGGCTGCAGAAGA-3′ and UBA reverse 5′-CTCGGGTAGCACGAGCTTTA-3′. All qRT–PCR experiments were performed in triplicates and the values represent means ± s.e.m.

### Histochemical analysis and microscopy

GUS staining was done as previously described (Péret *et al*, 2007). Plants were cleared for 24 h in 1 M chloral hydrate and 33% glycerol. Seedlings were mounted in 50% glycerol and observed with a Leica DMRB microscope. For confocal microscopy, plants were stained with 10 μg ml$^{-1}$ propidium iodide for 30 s and images were captured with an inverted confocal laser-scanning microscope (Leica TCS SP5 II).

Root cross-sections were reconstructed from 1 μm step z-series using the Volume viewer plugin for ImageJ.

## Auxin measurements

Roots were dissected and immersed into liquid medium containing 5 μM (Indole-D5)-IAM (C/D/N Isotopes Inc; Quebec, Canada) and incubated on gentle shaking. Samples containing ~10 mg plant material were collected after 10 and 20 h of incubation and frozen in liquid nitrogen. The samples were homogenised as previously described (Andersen *et al*, 2008), adding 250 pg (Acetic acid-D2)-IAA (C/D/N Isotopes Inc; Quebec, Canada) to each sample as an internal standard. Purification of the samples was done using 50 mg C18 BondElut SPE columns (Varian, Middelburg, The Netherlands) as described previously (Ljung *et al*, 2005). GC–MS/MS analysis was performed as described previously (Ljung *et al*, 2005) using the selected-reaction-monitoring (SRM) mode. Transitions from $m/z$ 261.1180 to $m/z$ 202.1050 for IAA, from $m/z$ 263.1309 to $m/z$ 204.1176 for the internal standard D2-IAA and from $m/z$ 266.1494 to $m/z$ 207.1361 for D5-IAA were recorded. The signal of D5-IAA was normalised against the signal of endogenous IAA.

## Statistical analysis

Statistical analyses for qPCR and auxin measurements were performed with Microsoft Excel on a personal computer. Asterisks indicate a significant difference with the relevant control experiment by Student's *t*-test (*$P<0.05$, **$P<0.01$ and ***$P<0.001$). The Hill function fits were computed using MATLAB routine fminsearch. Further details are provided in Supplementary Modelling Information. To calculate the frequencies of predicted LAX3 expression, we first computed the steady-state solutions to the model for various XPP cell auxin supply rates (Supplementary Modelling Figures M7–M25). For each model variant, we identified the level of auxin supply (by the primordium) at which levels of *LAX3* expression are half maximal. When the auxin supply is emanating from one XPP file, this was ~0.2, for three XPP files this was ~0.1. For this level of auxin supply, cells were said to express LAX3 (and were therefore counted) if their expression was above 5% of the maximal level. In the counting process, we distinguished between cells that are in the same file from those belonging to different files.

## Multicellular 3D modelling

Three versions of our model were developed; see Table I for a summary. Each of these was embedded in three-dimensional meshes using the pipeline described in Figure 4B–F. Upon generation, the 3D meshes were imported into the computer package MATLAB. Each variant of the model is described in detail in Supplementary Modelling Information, the models consisting of systems of coupled non-linear ODEs of the form:

$$\frac{d\boldsymbol{y}_i}{dt} = \boldsymbol{F}(\boldsymbol{y}_i; \boldsymbol{p}).$$

where $\boldsymbol{y}_i$ are the model variables (e.g., the level of IAA14 or auxin) for each cell $i$, the components of $\boldsymbol{p}$ are the model parameters (including rates of protein or mRNA decay, translation and rates of auxin transport) and those of $\boldsymbol{F}$ are the equations reflecting the interactions between the different system components. The models comprise systems of ODEs that describe the dynamics of the regulatory network controlling *LAX3* expression (the variants of which being depicted in Figures 4A, 5E and 7A). Thus, for each component $Z$ of the regulatory network (e.g., LAX3 or IAA14), we model how its level $z_i$ in cell $i$ is changes over time using ODEs of the form:

$$\frac{dz_i}{dt} = G_z(\boldsymbol{x}_i; \boldsymbol{q}) - H_z(\boldsymbol{x}_i; \boldsymbol{q}), \tag{1}$$

where $G_z$ is the synthesis rate of $Z$ and $H_z$ is its decay rate. Both of these are functions that depend on other components in the network, in addition to network-specific parameters $\boldsymbol{q}$ (e.g., rates of protein or mRNA decay and translation). Inputs in the network model (1) include the level

of auxin in a particular cell (this leading to the induction of downstream genes, including *LAX3*). Auxin levels are governed by a separate system of ODEs, which describe how auxin is transported throughout the 3D mesh (and so depend on network model, 1, outputs such as the level of LAX3, and additionally in model versions two and three, the level of PIN3 in a particular cell). The various geometrical parameters, including cell volume, surface area and cell–cell neighbour relations, were calculated directly from the particular 3D meshes. For the case where only LAX3 is present (and there is no PIN3, as in the case in model version one; see Supplementary Modelling Information for more information), the equations governing auxin transport take the form:

$$\frac{d[\text{auxin}_i]}{dt} = V_i^{-1}(P_{\text{IAAh}}(A_1 S_i[\text{auxin}^{\text{apo}}] - B_1 S_i[\text{auxin}_i]) + P_{\text{LAX3}}([\text{LAX3}_j])$$
$$\times (A_2 S_i[\text{auxin}^{\text{apo}}] - B_2 S_i[\text{auxin}_i]) + \omega_{\text{auxin}} - \mu_{\text{auxin}}[\text{auxin}_i],$$

where $[\text{auxin}_i]$ is the level of auxin in cell $i$ and $[\text{auxin}^{\text{apo}}]$ is the level of auxin in the adjoining apoplastic regions; $V_i^{-1}$ is the volume of cell $i$, $S_i$ is its total surface area; $P_{\text{LAAh}}$ and $P_{\text{LAX3}}$ are the membrane permeabilities associated with auxin diffusion and LAX3-mediated transport, respectively (noting the later is a function of LAX3 protein); $A_j$ and $B_j$ reflect the proportions of protonated auxin in the cytoplasm and apoplast, respectively (see Supplementary Modelling Information); $\omega_{\text{auxin}}$ and $\mu_{\text{auxin}}$ are the rates of auxin production and turnover, respectively. The full equations governing levels of auxin in the apoplast (namely $[\text{auxin}^{\text{apo}}]$) take a similar form, although here we take the limit where the apoplast thickness tends to 0 to obtain a quasi-steady state approximation (see Supplementary Modelling Information), and use approximation instead of solving a full system. This allows us to capture the effect of the apoplast without having to include an explicit compartment for it in the model. The governing equations for each of the model versions were solved using MATLAB routine ode15s. Model outputs were visualised using MEDIT (Frey, 2001).

## Supplementary information

## Acknowledgements

This work was supported by an FEBS Long-Term Fellowship (BP), an Intra-European Fellowship for Career Development under the 7th framework of the European Commission (IEF-2008-220506 to BP), an EMBO Long-Term Fellowship (BP), an European Reintegration Grant under the 7th framework of the European Commission (ERG-2010-276662 to BP) and the Swedish Research Council (VR 621-2010-5720 to IS, GS and KL). AMM, APF, AL, LRB, SP, NM, DMW, MO, JRK and MJB acknowledge the support of the Biotechnology and Biological Sciences Research Council (BBSRC) and Engineering and Physical Sciences Research Council (EPSRC) funding to the Centre for Plant Integrative Biology (CPIB); BBSRC Professorial Research Fellowship funding to DMW and MJB; Belgian Scientific policy (BELSPO contract MARS) to TB and MJB. We thank Bert de Rybel for his help in Multisite Gateway cloning.

*Author contributions:* BP and AMM contributed equally to this work. BP, AMM and MJB designed research. BP, AL, AB, MN, DMW, SP, IC, JK-V, SV, IS, RM, GS, KL, TB, EB, JF and IDS performed biology research. AMM, APF, LRB, EK, JRK, TP and MO performed mathematical/computational research. BP, AMM, JRK, MO and MJB analysed the data. BP, AMM and MJB wrote the paper. MJB is corresponding author.

## Conflict of interest

The authors declare that they have no conflict of interest.

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
