## [Review Process File · Molecular Systems Biology]

Sequential induction of auxin efflux and influx carriers regulates lateral root emergence

Benjamin Péret, Alistair M Middleton, Andrew P French, Antoine Larrieu, Anthony Bishopp, Maria Njo, Darren M Wells, Silvana Porco, Nathan Mellor, Leah R Band, Ilda Casimiro, Jurgen Kleine-Vehn, Steffen Vanneste, Ilkka Sairanen, Romain Mallet, Goran Sandberg, Karin Ljung, Tom Beeckman, Eva Benkova, Jiri Friml, Eric Kramer, John R King, Ive De Smet, Tony Pridmore, Markus Owen and Malcolm J Bennett

Corresponding author: Malcolm Bennett, University of Nottingham, UK

Review timeline:

Submission date:	30 January 2012
Editorial Decision:	05 March 2012
Resubmission:	15 March 2013
Editorial Decision:	17 May 2013
Revision received:	03 July 2013
Accepted:	05 July 2013

Editor: Maria Polychronidou

Transaction Report:

1st Editorial Decision

05 March 2012

Thank you again for submitting your work to Molecular Systems Biology. We have now heard back from the three referees who agreed to evaluate your manuscript. As you will see from the reports below, the referees raise substantial concerns on your work, which, I am afraid to say, preclude its publication.

While the reviewers expressed some interest in this work, they had fundamental concerns regarding both the conclusiveness of the experimental evidence, and the biological appropriateness of aspects of the model-based analyses. Both the first and last reviewer felt that substantial additional experimental evidence and new modeling work would be needed to support the main claims made in this work. The reviewers also shared important concerns regarding the presentation of this work, feeling that important details were lacking and many key experiments were presented in the supplementary material, which, overall, indicates that this work would not be appropriate for our Report format.

Given these rather extensive concerns, we feel we have no choice but to return this manuscript with the message that we cannot offer to publish it.

Nevertheless, the reviewers did have positive words for the goals of this study, and all felt that a substantially expanded work, with new experiments (including important controls), additional modeling work, and a more thorough presentation might be suitable for Molecular Systems Biology.

As such, we would like to indicate that we may be willing to reconsider a new submission based on this work. Any new submission would need to include new experimental data rigorously addressing the concerns raised by the first and last reviewers, as well as consideration of the developmental changes in 3D structure during lateral root emergence, which the last reviewer felt would be essential to include in the model. A resubmitted work should be formatted as a full-length Article, rather than a Report.

A resubmitted work would have a new number and receipt date. We recognise that this would involve substantial additional experimentation and analysis, and, as you probably understand, we can give no guarantee about its eventual acceptability. If you do decide to follow this course then it would be helpful to enclose with your re-submission an account of how the work has been altered in response to the points raised in the present review.

I am sorry that the review of your work did not result in a more favorable outcome on this occasion, but I hope that you will not be discouraged from sending your work to Molecular Systems Biology in the future.

Thank you for the opportunity to examine this work.

REFEREE REPORTS

Reviewer #1

Review of the manuscript: "Lateral root emergence depends on an inductive signal, tissue geometry and gene"

The authors submit a manuscript that combines some experimental work with modelling to explain the specific expression pattern of the LAX3 gene of Arabidopsis during lateral root emergence. Simply put, it looks like an auxin highway from the central xylem pole pericycle cell through overlaying cortex cells determines the expression pattern, which based on previous work guides lateral root emergence through the overlaying tissues. I have a number of comments on this manuscript as outlined below, some of them general, some concerning experimental procedures, some specific to the modelling part.

1. To start with, I believe this story could be presented more comprehensively and coherently. It seems to me that this manuscript had initially been submitted as a short letter type article to other journals and has only been minimally adapted for publication in MSB. I think the authors should consider expanding the manuscript in all sections and moving some of the supplemental figures into the main manuscript so it can be largely understood as a stand alone paper, w/o necessary reference to suppl. data for the general reader.
2. In places the manuscript is assembled sloppily, for example there are many discrepancies such as upper versus lower case figure labeling, Supp. Figure 8 is cited after Supp. fig. 9, etc., etc. The authors should go through the manuscript carefully to fix these mistakes.
3. The central observation that serves as a starting point of this paper is the specific expression of LAX3 in two cortex cells during lateral root emergence. I think it is important that the authors provide a clear picture of the stochasticity of this process. Is it really always two cells? What statistical confidence can we put in this finding? How many lateral root primordia have been observed, how often was LAX3 expression observed in one, two, three or even four cells? Is this robust throughout the emergence process? I believe statistically solid data for this are absolutely necessary to decide whether the subsequent modelling is even justified rather than futile.
4. Overall, statistical treatment of the data reveals borderline significance in various places. For some data (e.g. the qPCR data), large relative differences are accompanied by rather weak (<0.05) p values, pointing to a strong effect of outliers. I think the authors have to make a more convincing effort and indicate number of samples/observations and the real p values (maybe p is <0.01 or <0.001 in places, but only 0.05 was considered).

5. I think it is important to show proper controls for all experiments. For instance, I would like to see an optical cross section of J021>>GFP to demonstrate its pericycle xylem pole-specific expression, rather than merely a suggestive drawing (Fig. 1L).
6. Regarding LAX3 expression analysis, it is not clear from the figure legends or methods what we're actually looking at. This is also because the construct nomenclature in the methods is not standard (e.g. LAX3YFP is LAX3 promoter driving YFP coding, LAX3 YFP fusion protein {driven by which promoter?} etc. With respect to this, figure legends are much too minimalist and should be expanded.
7. Further regarding LAX3 expression, my impression is that *iaaH* induction also triggers LAX3 expression in the epidermis. Is this correct (Fig. 1M)? By contrast, I do not see expansion of LAX3 expression in Fig. 3N as stated in the text. Is this a dosage effect of IAM treatment or is the image simply misleading? Moreover, what is the signal from the stele? Is it background fluorescence from lignified cells or true LAX3 expression? If so, this should be pointed out in the text.
8. Still regarding LAX3 expression, a mock control for Fig. 1 M & N should be shown.
9. Regarding the model, the authors simulate the LAX3 gene regulatory network within a realistic 3D section of the root zone where lateral root emergence takes place. It is not clear from the manuscript which is the added value of using a third dimension in the 3D model instead of adopting, for instance, a simpler 2D representation of a radial section of the root similar to the one that is depicted in the figures of the manuscript where the simulation results are shown. There is also a discrepancy between the main text and the model description. It is not clear whether a single cross section was used and cell length was simulated, or whether indeed several consecutive sections were used to generate the 3D model.
10. As far as I see, the authors do not specify if the model takes into account the rootward polar auxin transport through the vasculature and the shootward polar auxin transport through the epidermal layers.
11. The inner part of the root does not seem to be explicitly modelled, since the focus is only on the xylem pole pericycle cell file as the source of auxin. In this way, the authors are not considering the effects that the higher auxin concentration present in other vascular tissues would have on the outer cell layers. I think the authors have to present evidence or explain comprehensively why this reductionist approach would be justified.
12. In Figure 3D, the authors refer to the induction of LAX3-YFP under the application of different concentration of auxin as a sigmoidal response. The dose-response curve shown in the plot looks different from a typical sigmoidal curve. It is clear for instance that a steep decrease follows the induction. Regarding the sigmoid induction across shorter time as shown in the supplemental data, the fit is somewhat minimalist, since there are only three real data points. I think more data points are needed to clearly see whether the claimed sigmoid behavior is actually real.
13. In Figure 2A-D, the auxin influx carrier LAX3 and the auxin efflux carrier indicated as AEC have both a positive effect on auxin. If the gene regulatory network is represented here at the intracellular level, then LAX3 and the AEC should have opposite effects on auxin due to their carrier properties, namely a positive effect on intracellular auxin of the LAX3 auxin influx carrier and a negative effect on the intracellular auxin resulting from the efflux action of the AEC carrier. I think this is an important logical discrepancy that has to be fixed. Is the model still giving the same outcome if the relation between AEC and IAA is negative instead of positive as would be conceptually correct?
14. The authors emphasize the importance of tissue geometry for their model, but I do not see that this has been explicitly tested. Can the authors provide direct evidence for its importance? Can they for example manipulate their model and create a version in which cortex cell sizes are changed and contact areas are consequently more equal or smaller for the two central cells and then test for the outcome?
15. Finally, the citations are somewhat self-centered. The authors should acknowledge the work of

others who have made important modelling contributions to our understanding of auxin transport, lateral root positioning and specific expression patterning.

Reviewer #2

This paper addresses an interesting problem in the context of lateral root formation. Via an experiment-theory loop the authors offer a consistent explanation for the locally confined expression pattern of LAX3. For this they use a multi-scale model into which the relevant system properties are integrated. The model predicted several features (e.g. the existence of an auxin efflux carrier), which the authors verified experimentally. This manuscript is an excellent example for a project in which experiment AND theory are necessary to gain new and deeper insight into the behavior of a system. I consider the manuscript as absolutely appropriate for publication in Molecular Systems Biology.

However, I have two problems with the paper in its current form.

1) Presentation of the results. It is very difficult to understand what the authors - in particular the mathematical part - have really done. Phrases like: 'analysis of the model indicated ...' sound a bit like: 'we have done something useful, but please figure out yourself what it was'. I had to heavily refer to the supplement to understand what the model is, what assumptions go into it, etc. Without these information how should a reader assess the results the authors present? Another indication that the amount of information in the main text and in the supplement is not balanced, is that the authors refer more often to figures in the supplement than to figures in the main text. For me the strongest finding of the paper is the prediction and experimental verification of the difference between the response time of LAX3 and PIN3. Again, it is not possible to understand why the model predicts it, without reading the supplement.

In my opinion the main text of a paper should be sufficient to understand what the authors have done; this should hold for both the experimental part AND the theoretical/mathematical part. Therefore, I request that the authors improve this in their manuscript and avoid phrases as 'model analysis indicated' and rather explain why and how the analysis revealed something. The model assumptions should become clear in the main text. For the experimental part no one would accept a pseudo-presentation like: 'we have done some experiments and they reveal ... (if you want to know which experiments and how they were done, please read the supplement)', so it has to equally hold for the theoretical part. I also suggest that the very interesting Figure S7A goes into the main text.

2) How much does the results, in particular the confinement of the LAX3 expression to only two cell files, depend on the geometry of the grid? Why is it that in Figure 2A the expression level of LAX3 is lowest in the lowest of the three LAX3 expressing cells? And could it be that a change of cell shapes would result again in three cells expression LAX3? And what does the term: 'careful examination ...' on line 91 really mean? Is LAX3 expression always (with no exception) confined to these two cell files, as the authors indicate in the text? Since I assume that the cell shapes will always be slightly different from root to root this would mean a robust feature of the system (i.e. insensitive to cell shape). Is this reflected as well in the mathematical model?

To summarize, I believe this work is excellent and is well suited for publication in Molecular Systems Biology if the authors answer my last question and rework their presentation of their work.

Reviewer #3

The paper by Péret, Middleton et al deals with understanding how expression of genes necessary for emergence of lateral roots can be determined spatially by the hormone auxin. Auxin can be mobilized by the activity of influx and efflux carriers and used to activate, in turn, specific expression of one of these carriers (LAX3) as well as cell wall separation enzymes, which are required for lateral root emergence. Auxin transport has been modeled previously but not in the context of lateral root emergence. In addition, this paper includes modeling in 3D which is relevant to further understand how specific expression patterns can be generated in space. The paper is novel and the modeling should help to unravel how a complex developmental mechanism such as lateral root emergence is facilitated by specific gene expression in space. However, there are a number of issues that require that the current version is completely revised:

1) The authors model a developmental stage where expression of the gene LAX3 in adjacent layers to the lateral root primordium (LRP), if it occurs, does not seem to be relevant to its function. Based on publications from this group (Swarup et al, 2008) or this paper (Fig 1A and 1B) expression of the gene LAX3 in adjacent layers to the LRP can be detected as early as in stage II of lateral root development and continues throughout subsequent development -to emergence. In addition, phenotypes related to the *lax3* loss of function mutant indicate a role for this gene starting at stage II (or maybe stage I). However, the authors model an earlier developmental stage than stage I. The models are developed for pericycle cells that do not show any morphological change when compared to other pericycle cells. As one of the main strengths of this paper is the 3D modeling, this modeling should account for the spatial changes associated to LRPs, at least at stage II, and that subsequently could explain the specific expression of LAX3 in adjacent cell layers (and maybe its absence from dividing pericycle cells). The current models are oversimplified to the point that they ignore the spatial conformation of the LRP at the appropriate development stage. As a result and based on published information or in this paper (Fig 1A, 1B), these models (in their current state) do not appear to be suitable to explain/infer expression patterns or regulatory connections with a biological relevance for the studied developmental mechanism.

2) Xylem pole pericycle cells are the cells in contact with xylem cells. In figure S1 there are two cells in contact with xylem (protoxylem) on the right side/pole of the root. Therefore, based on this criterion there are 2 xylem pole pericycle cells not three as the authors say. It is possible that there are 3 cells in contact with xylem in other cases. For instance, in the same root shown in figure S1 there appear to be 3 xylem pole pericycle cells at the other pole (left side). As mentioned before, using pericycle cells in the models does not seem appropriate but it might be informative in combination with anatomical studies. These anatomical studies should reveal the morphology of a stage II LRP. It will be interesting to learn if stage II LRPs comprise 2 or 3 files of dividing 'pericycle' cells or maybe both cases can be found. Based on this information, the authors could then decide what spatial organization best represents a LRP at stage II of development and use it to develop the required models.

3) The relationship between the presence of auxin in the pericycle and expression of LAX3 in adjacent cell layers should be put in the context of LRPs. The authors have developed a very interesting system to see how the presence of auxin in pericycle cells affects expression of LAX3 in adjacent cell layers located towards the outside of the root. The authors should look more closely at changes occurring after application of D5-IAM to determine the relationship between red-tagged cells (*iaaH-RFP*), which should produce auxin, and yellow-tagged cells (*LAX3-YFP*), which express the gene of interest. This information is missing in the paper and more specifically they authors should address:

3a) It is unclear if *iaaH-RFP* is indeed expressed in three files of cells at the xylem pole as the authors assume or rather in two files as one would expect from the previously observed expression pattern of J0121. The authors cannot conclude that 3 files of cells are not needed if they do not know if their construct is actually expressed in these 3 files of cells. If this was the case, this reviewer feels that it is also important to clarify that the authors could only conclude that the three files cells are not required or cannot recapitulate the observed expression pattern of LAX3. It cannot be concluded that only one file of cells is required (as alternative explanation to three file of cells are required) as there are other possibilities (i.e. a negative signal coming from any of these files of cells or even two files of cells are required). Therefore, if the authors want to conclude that only one file of pericycle cells is required (if relevant, see 1) they would need to empirically test it.

3b) The authors appear to assume that pericycle cells do not undergo any morphological change (i.e. cell division) after D5-IAM application. However, cell division is likely to occur in these cells and it appears to be the case in Fig 1M and 1N. Fig 1N is especially confusing as there seems to be a LRP on the right side and only two cortical cells expressing LAX3 (at that side) but not 3 as the authors say. In addition, Fig 1L does not really summarize what it is shown in the other pictures.

Do the authors see expression of LAX3 in all cortical cells adjacent to xylem pole pericycle cells after D5-IAM application or only in the ones close to pericycle cells that undergo division (if they do)? What is the earliest time they see expression of LAX3 in cortical cells? 10 or more hours as one may infer from Fig S2E? This section would greatly benefit from a time course experiment where

the authors related expression of *iaaH*-RFP to expression of *LAX3* and *D5-IAA* levels in roots.

Based on the current results the authors cannot really conclude that auxin coming from xylem pole pericycle cells (instead of divided pericycle cells or LRP) activates expression of *LAX3*. The authors should determine first the relationship between *LAX3* cortical expression and divided/undivided pericycle cells after *D5-IAM* application.

4) The title and abstract should be aimed for a broader audience.

5) There are several concepts that need to be explained better and put in a developmental context (auxin, auxin-carrier, auxin transport...). Some other concepts should be used more precisely. For instance, the authors refer to 'the signal auxin' or 'the inductive auxin signal', which might be appropriate in a Physics paper, but in this paper it should be clearer that auxin is a plant hormone required for morphogenetic process and that it provides positional information. Mentioning '*LAX3* regulatory network' is also confusing as *LAX3* is not a transcriptional regulator but an auxin carrier. Finally, the use of the words 'source of auxin' to refer to LRPs is misleading as it is not known if auxin is being produced in the LRPs or just passes through them -coming from vascular tissues.

6) It is not clear why the models do not predict expression of *LAX3* in the endodermis regardless of using LRPs in the model or not.

7) Fig S10, it is unclear why the addition of the percentages of all the bars in the graph is greater than 100%

Resubmission

15 March 2013

Reviewer #1

Review of the manuscript: "Lateral root emergence depends on an inductive signal, tissue geometry and gene network topology"

*The authors submit a manuscript that combines some experimental work with modelling to explain the specific expression pattern of the *LAX3* gene of *Arabidopsis* during lateral root emergence. Simply put, it looks like an auxin highway from the central xylem pole pericycle cell through overlaying cortex cells determines the expression pattern, which based on previous work guides lateral root emergence through the overlaying tissues. I have a number of comments on this manuscript as outlined below, some of them general, some concerning experimental procedures, some specific to the modelling part.*

1. To start with, I believe this story could be presented more comprehensively and coherently. It seems to me that this manuscript had initially been submitted as a short letter type article to other journals and has only been minimally adapted for publication in MSB. I think the authors should consider expanding the manuscript in all sections and moving some of the supplemental figures into the main manuscript so it can be largely understood as a stand alone paper, w/o necessary reference to suppl. data for the general reader.

As suggested by reviewer 1, we have significantly expanded the manuscript (into an Article format) and moved several supplementary figures into the main text so that we now have 7 main figures.

2. In places the manuscript is assembled sloppily, for example there are many discrepancies such as upper versus lower case figure labeling, Supp. Figure 8 is cited after Supp. fig. 9, etc., etc. The authors should go through the manuscript carefully to fix these mistakes.

The revised manuscript has been thoroughly edited to remove these issues.

*3. The central observation that serves as a starting point of this paper is the specific expression of *LAX3* in two cortex cells during lateral root emergence. I think it is important that the authors provide a clear picture of the stochasticity of this process. Is it really always two cells? What*

statistical confidence can we put in this finding? How many lateral root primordia have been observed, how often was LAX3 expression observed in one, two, three or even four cells? Is this robust throughout the emergence process? I believe statistically solid data for this are absolutely necessary to decide whether the subsequent modelling is even justified rather than futile.

As suggested, we performed a statistical analysis of LAX3 expression pattern. This approach confirms the observed pattern and the corresponding results have been included in Figure 1.

4. Overall, statistical treatment of the data reveals borderline significance in various places. For some data (e.g. the qPCR data), large relative differences are accompanied by rather weak (<0.05) p values, pointing to a strong effect of outliers. I think the authors have to make a more convincing effort and indicate number of samples/observations and the real p values (maybe p is <0.01 or <0.001 in places, but only 0.05 was considered).

Weak p values correspond to the small relative induction of LAX3 in the *pin2* mutant or in response to IAM treatment. These small, yet significant p values, are the result of LAX3 being strongly expressed in the stele (as seen in all confocal cross sections). Therefore the induction of LAX3 in the outer cells is low compared to its basal level. However, these results are independently validated by promoter:GUS/GFP approaches and are therefore very reliable. All other qPCR show strong induction level as reflected by stronger p values.

5. I think it is important to show proper controls for all experiments. For instance, I would like to see an optical cross section of J021>>GFP to demonstrate its pericycle xylem pole-specific expression, rather than merely a suggestive drawing (Fig. 1L).

The expression pattern of the J0121 line is well documented (Laplaze L, et al., 2005 *J Exp Bot* 56: 2433-42 and Parizot B, et al., 2008 *Plant Physiol* 146: 140-8), however we have included a cross section showing its GFP expression pattern in the 3 files of xylem pole pericycle cells (Figure 3I) as previously reported.

6. Regarding LAX3 expression analysis, it is not clear from the figure legends or methods what we're actually looking at. This is also because the construct nomenclature in the methods is not standard (e.g. LAX3YFP is LAX3 promoter driving YFP coding, LAX3 YFP fusion protein {driven by which promoter?} etc. With respect to this, figure legends are much too minimalist and should be expanded.

The LAX3-YFP line has been described previously (Swarup et al., 2008). However, we have updated its description both in the text and the figure legend to make it clear to readers that it is 'a functional pLAX3:LAX3YFP fusion' that has been used throughout our study.

*7. Further regarding LAX3 expression, my impression is that *iaaH* induction also triggers LAX3 expression in the epidermis. Is this correct (Fig. 1M)? By contrast, I do not see expansion of LAX3 expression in Fig. 3N as stated in the text. Is this a dosage effect of IAM treatment or is the image simply misleading? Moreover, what is the signal from the stele? Is it background fluorescence from lignified cells or true LAX3 expression? If so, this should be pointed out in the text.*

LAX3 is indeed expressed in the epidermis during LR emergence at later time points (as described previously in Swarup et al., 2008) and the IAM treatment induces expression in the epidermis accordingly. However, the focus of the current study is on LAX3 induction in the cortex during LR initiation and do not use 3D geometries from later stages of LRP development, hence we can only infer that LAX3 induction in the epidermis follows the same pattern and rules as in the cortex.

Expansion of LAX3 expression is clearly visible in Figure 3N (now Figure 3L) as 5 cortical cells are now expressing LAX3 instead of 2 for the untreated control.

As noted by the reviewer and reported previously (Swarup et al., 2008), pLAX3:LAX3YFP is strongly expressed in the stele, but we showed previously that this expression pattern does not impact LAX3 role in LR emergence.

8. Still regarding LAX3 expression, a mock control for Fig. 1 M & N should be shown.

As suggested, we included a mock control for Figure 1 M and N (now Figure 3 K and L). This control is in Fig. 3J.

9. Regarding the model, the authors simulate the LAX3 gene regulatory network within a realistic 3D section of the root zone where lateral root emergence takes place. It is not clear from the manuscript which is the added value of using a third dimension in the 3D model instead of adopting, for instance, a simpler 2D representation of a radial section of the root similar to the one that is depicted in the figures of the manuscript where the simulation results are shown. There is also a discrepancy between the main text and the model description. It is not clear whether a single cross section was used and cell length was simulated, or whether indeed several consecutive sections were used to generate the 3D model.

As our statistical analysis revealed, the expression pattern of pLAX3:LAX3YFP is three-dimensional (number of cell files and number of cells in each file). Also, auxin originating from the LRP will be transported not only radially but also longitudinally. As a result, we anticipated that a 3D model would be more appropriate. We therefore used a cross section to obtain the 2D matrix and the longitudinal axis was simulated. Cells walls were added in the longitudinal dimension to match experimentally observed cell lengths and organisation (i.e. cells in different files are arranged in a staggered formation). As a result, we obtained a realistic 3D matrix. Using this we found that the final iteration of the model could account for the stereotypical LAX3 expression pattern, namely that it is expressed in a total of two-four cells, with two radially and two longitudinally.

10. As far as I see, the authors do not specify if the model takes into account the rootward polar auxin transport through the vasculature and the shootward polar auxin transport through the epidermal layers.

Our model takes into account auxin that accumulates in the XPP cell and then is transported towards the outer tissue. It is known that auxin comes from the underlying vasculature and before that from the leaves as demonstrated previously (Swarup et al., 2008). However, the ability for auxin to accumulate or not in xylem-pole cells is a different developmental process (i.e, vascular patterning) that is beyond the scope of this current manuscript and is controlled by an entirely different gene regulatory network (see Bishopp et al, 2011 in Current Biology 21:11, 917-26). Here we only considered XPP cells that are accumulating auxin (i.e, right after LR initiation has occurred). Concerning the shootward auxin transport in the epidermis, this auxin route is restricted to the young part of the root close to the meristem as described in the literature (see Petrásek J and Friml J 2009 *Development* 136, 2675-88 for a recent review). Therefore this mode of transport does not operate in the zone where LAX3 is expressed (mature part of the root, away from the apical meristem). The rootward and shootward polar auxin transports are therefore not relevant to the present study.

11. The inner part of the root does not seem to be explicitly modelled, since the focus is only on the xylem pole pericycle cell file as the source of auxin. In this way, the authors are not considering the effects that the higher auxin concentration present in other vascular tissues would have on the outer cell layers. I think the authors have to present evidence or explain comprehensively why this reductionist approach would be justified.

We have modelled the auxin distribution in the inner part of the root in a separate study (Muraro et al, under review in PNAS) using a 2D +t model of a root cross-section. Our model predicted that auxin accumulates in xylem pole PLUS overlaying xylem pole pericycle cells (which was experimentally validated using the auxin-responsive reporter AHP6:GFP).

However, this current study focusses on XPP cells that have already undergone LR initiation and are therefore considered as founder cells, meaning that they have accumulated auxin. This is therefore the starting point of our model that aims at understanding how auxin is then transported towards the outer tissue to induce LAX3 and control its expression pattern.

12. In Figure 3D, the authors refer to the induction of LAX3-YFP under the application of different

concentration of auxin as a sigmoidal response. The dose-response curve shown in the plot looks different from a typical sigmoidal curve. It is clear for instance that a steep decrease follows the induction. Regarding the sigmoid induction across shorter time as shown in the supplemental data, the fit is somewhat minimalist, since there are only three real data points. I think more data points are needed to clearly see whether the claimed sigmoid behavior is actually real.

As suggested, we performed a more detailed dose response analysis (8 concentrations were used). This allowed us to confirm the sigmoidal behaviour of LAX3 induction by auxin (Fig. 4H).

13. In Figure 2A-D, the auxin influx carrier LAX3 and the auxin efflux carrier indicated as AEC have both a positive effect on auxin. If the gene regulatory network is represented here at the intracellular level, then LAX3 and the AEC should have opposite effects on auxin due to their carrier properties, namely a positive effect on intracellular auxin of the LAX3 auxin influx carrier and a negative effect on the intracellular auxin resulting from the efflux action of the AEC carrier. I think this is an important logical discrepancy that has to be fixed. Is the model still giving the same outcome if the relation between AEC and IAA is negative instead of positive as would be conceptionally correct?

We apologize for this mistake in the cartoon and it has been corrected (see new Fig. 5D and E). However, the original model indeed took into account that the auxin efflux carrier removes auxin from the cell and therefore has a negative effect on auxin accumulation.

14. The authors emphasize the importance of tissue geometry for their model, but I do not see that this has been explicitly tested. Can the authors provide direct evidence for its importance? Can they for example manipulate their model and create a version in which cortex cell sizes are changed and contact areas are consequently more equal or smaller for the two central cells and then test for the outcome?

To explore this valid point, we have used additional root cross sections to generate several different tissue geometries. Surprisingly, we observed that in the first version of the model (which lacks AEC/PIN3), LAX3 expression pattern is very sensitive to variation in cell geometry. However, when we include PIN3, the model becomes robust, such that the wildtype LAX3 spatial expression pattern can be captured in every root geometry we tested.

15. Finally, the citations are somewhat self-centered. The authors should acknowledge the work of others who have made important modelling contributions to our understanding of auxin transport, lateral root positioning and specific expression patterning.

We have cited several papers from other groups who modelled auxin fluxes during primary root (e.g. Grienenen et al, 2007) and lateral root formation (e.g. Laskowski et al, 2008).

Reviewer #2

This paper addresses an interesting problem in the context of lateral root formation. Via an experiment-theory loop the authors offer a consistent explanation for the locally confined expression pattern of LAX3. For this they use a multi-scale model into which the relevant system properties are integrated. The model predicted several features (e.g. the existence of an auxin efflux carrier), which the authors verified experimentally. This manuscript is an excellent example for a project in which experiment AND theory are necessary to gain new and deeper insight into the behavior of a system. I consider the manuscript as absolutely appropriate for publication in Molecular Systems Biology.

However, I have two problems with the paper in its current form.

1) *Presentation of the results. It is very difficult to understand what the authors - in particular the mathematical part - have really done. Phrases like: 'analysis of the model indicated ...' sound a bit like: 'we have done something useful, but please figure out yourself what it was'. I had to heavily refer to the supplement to understand what the model is, what assumptions go into it, etc. Without*

these information how should a reader assess the results the authors present? Another indication that the amount of information in the main text and in the supplement is not balanced, is that the authors refer more often to figures in the supplement than to figures in the main text. For me the strongest finding of the paper is the prediction and experimental verification of the difference between the response time of LAX3 and PIN3. Again, it is not possible to understand why the model predicts it, without reading the supplement.

In my opinion the main text of a paper should be sufficient to understand what the authors have done; this should hold for both the experimental part AND the theoretical/mathematical part. Therefore, I request that the authors improve this in their manuscript and avoid phrases as 'model analysis indicated' and rather explain why and how the analysis revealed something. The model assumptions should become clear in the main text. For the experimental part no one would accept a pseudo-presentation like: 'we have done some experiments and they reveal ... (if you want to know which experiments and how they were done, please read the supplement)', so it has to equally hold for the theoretical part. I also suggest that the very interesting Figure S7A goes into the main text.

In response to reviewers valid comments, our manuscript has been substantially enlarged with a more thorough presentation formatted (as requested) as a full-length Article, rather than a Report, enabling us to provide more details about the modelling in the expanded text. Also, Figure S7A now appears as a main text figure (now Figure 6A).

2) How much does the results, in particular the confinement of the LAX3 expression to only two cell files, depend on the geometry of the grid? Why is it that in Figure 2A the expression level of LAX3 is lowest in the lowest of the three LAX3 expressing cells? And could it be that a change of cell shapes would result again in three cells expression LAX3? And what does the term: 'careful examination ...' on line 91 really mean? Is LAX3 expression always (with no exception) confined to these two cell files, as the authors indicate in the text? Since I assume that the cell shapes will always be slightly different from root to root this would mean a robust feature of the system (i.e. insensitive to cell shape). Is this reflected as well in the mathematical model?

To address this valid point we generated several different tissue geometries. Surprisingly, we have found that in the first version of the model (which lacks AEC/PIN3), the expression of LAX3 is very sensitive to natural variations in geometry. However, when we include PIN3, the model is extremely robust, so that the typical wildtype pattern can be captured in every root geometry we tested.

To summarize, I believe this work is excellent and is well suited for publication in *Molecular Systems Biology* if the authors answer my last question and rework their presentation of their work.

We appreciate reviewer 2's support and encouragement

Reviewer #3

The paper by Péret, Middleton et al deals with understanding how expression of genes necessary for emergence of lateral roots can be determined spatially by the hormone auxin. Auxin can be mobilized by the activity of influx and efflux carriers and used to activate, in turn, specific expression of one of these carriers (LAX3) as well as cell wall separation enzymes, which are required for lateral root emergence. Auxin transport has been modeled previously but not in the context of lateral root emergence. In addition, this paper includes modeling in 3D which is relevant to further understand how specific expression patterns can be generated in space. The paper is novel and the modeling should help to unravel how a complex developmental mechanism such as lateral root emergence is facilitated by specific gene expression in space. However, there are a number of issues that require that the current version is completely revised:

1) The authors model a developmental stage where expression of the gene LAX3 in adjacent layers to the lateral root primordium (LRP), if it occurs, does not seem to be relevant to its function. Based on publications from this group (Swarup et al, 2008) or this paper (Fig 1A and 1B) expression of the gene LAX3 in adjacent layers to the LRP can be detected as early as in stage II of lateral root development and continues throughout subsequent development -to emergence. In addition, phenotypes related to the *lax3* loss of function mutant indicate a role for this gene starting at stage

II (or maybe stage I). However, the authors model an earlier developmental stage than stage I. The models are developed for pericycle cells that do not show any morphological change when compared to other pericycle cells. As one of the main strengths of this paper is the 3D modeling, this modeling should account for the spatial changes associated to LRPs, at least at stage II, and that subsequently could explain the specific expression of LAX3 in adjacent cell layers (and maybe its absence from dividing pericycle cells). The current models are oversimplified to the point that they ignore the spatial conformation of the LRP at the appropriate development stage. As a result and based on published information or in this paper (Fig 1A, 1B), these models (in their current state) do not appear to be suitable to explain/infer expression patterns or regulatory connections with a biological relevance for the studied developmental mechanism.

As shown in Figure 1 C and D, LAX3 expression in 2 cortical cells is seen as soon as stage I, which is in accordance with accumulation of Stage I LRP in the *lax3* mutant (Swarup et al., 2008). We therefore strongly disagree with the statement that the spatial changes in the LRP are important as the changes in shape at Stage I are negligible as commonly reported in the literature (Malamy and Benfey, 1997). This is the reason why we choosed to model the early steps of LR emergence, from the moment the XPP cell starts to accumulate auxin until the induction of LAX3 in the cortical cells (corresponding to ca. 6 to 12 hours). Moreover, the model has been able to generate predictions that have subsequently been tested experimentally, and so undergone several iterations in the paper. This has led to the identification of new components (PIN3) and predictions on their dynamics (that LAX3 and PIN3 are induced consecutively). Together this strongly argues that the model does have a biological relevance.

2) Xylem pole pericycle cells are the cells in contact with xylem cells. In figure S1 there are two cells in contact with xylem (protoxylem) on the right side/pole of the root. Therefore, based on this criterion there are 2 xylem pole pericycle cells not three as the authors say. It is possible that there are 3 cells in contact with xylem in other cases. For instance, in the same root shown in figure S1 there appear to be 3 xylem pole pericycle cells at the other pole (left side). As mentioned before, using pericycle cells in the models does not seem appropriate but it might be informative in combination with anatomical studies. These anatomical studies should reveal the morphology of a stage II LRP. It will be interesting to learn if stage II LRPs comprise 2 or 3 files of dividing 'pericycle' cells or maybe both cases can be found. Based on this information, the authors could then decide what spatial organization best represents a LRP at stage II of development and use it to develop the required models.

We don't agree with the definition of XPP cells. They are not the cells in contact with the xylem, but the cells situated in front of it. For a commonly observed number of 14 pericycle cells, this defines 2 phloem poles of 4 cells and 2 xylem poles of 3 cells. This fits with previous observation that LRP are derived from 3 cells of XPP cells but that the central file mainly contributes to the LRP formation whereas the 2 side files contribute only to the formation of LRP flanks (Kurup S et al., 2005 Plant J 42: 444-53). More recent work from our laboratory has shown that this was true on several hundreds of LRP (Lucas M et al., 2013 PNAS in press).

*3) The relationship between the presence of auxin in the pericycle and expression of LAX3 in adjacent cell layers should be put in the context of LRPs. The authors have developed a very interesting system to see how the presence of auxin in pericycle cells affects expression of LAX3 in adjacent cell layers located towards the outside of the root. The authors should look more closely at changes occurring after application of D5-IAM to determine the relationship between red-tagged cells (*iaaH-RFP*), which should produce auxin, and yellow-tagged cells (*LAX3-YFP*), which express the gene of interest.*

Benjamin ?

This is an interesting point but, as stated above, morphological changes are outside the focus of the paper (which addresses the mechanisms regulating the induction of LAX3 expression).

More specifically they authors should address:

3a) It is unclear if *iaaH-RFP* is indeed expressed in three files of cells at the xylem pole as the authors assume or rather in two files as one would expect from the previously observed expression pattern of J0121. The authors cannot conclude that 3 files of cells are not needed if they do not know if their construct is actually expressed in these 3 files of cells. If this was the case, this reviewer feels that it is also important to clarify that the authors could only conclude that the three files cells are not required or cannot recapitulate the observed expression pattern of *LAX3*. It cannot be concluded that only one file of cells is required (as alternative explanation to three file of cells are required) as there are other possibilities (i.e. a negative signal coming from any of these files of cells or even two files of cells are required). Therefore, if the authors want to conclude that only one file of pericycle cells is required (if relevant, see 1) they would need to empirically test it.

In accordance to our comment above, the J0121 line that specifically drives expression in XPP cells clearly shows 3 cell files (see Figure 2G plus Laplaze et al., 2005 *J Exp Bot* 56: 2433-42 and Parizot et al., 2008 *Plant Physiol* 146: 140-8). However, we agree that targeted auxin synthesis in these files is not sufficient to conclude that only 1 cell is required. We therefore included observations that the DR5:GUS auxin response reporter is stronger in the central XPP cell (Fig. 5 J-L).

3b) The authors appear to assume that pericycle cells do not undergo any morphological change (i.e. cell division) after D5-IAM application. However, cell division is likely to occur in these cells and it appears to be the case in Fig 1M and 1N. Fig 1N is especially confusing as there seems to be a LRP on the right side and only two cortical cells expressing *LAX3* (at that side) but not 3 as the authors say. In addition, Fig 1L does not really summarize what it is shown in the other pictures.

Periclinal cell division is not relevant to the present study as it represents an event happening more than 12 hours after auxin application at a time where *LAX3* expression can already be observed.

*Do the authors see expression of *LAX3* in all cortical cells adjacent to xylem pole pericycle cells after D5-IAM application or only in the ones close to pericycle cells that undergo division (if they do)? What is the earliest time they see expression of *LAX3* in cortical cells? 10 or more hours as one may infer from Fig S2E? This section would greatly benefit from a time course experiment where the authors related expression of *iaaH-RFP* to expression of *LAX3* and D5-IAA levels in roots.*

We performed a detailed time course experiment (Fig. 6E) showing that exogenous auxin application triggers *LAX3* induction after 3-4 hours. Labelling of the auxin precursor IAM demonstrates that labelled auxin is detected as early as 10 hours after application. This coincides with the observation that *LAX3* is expressed in front of stage I LRP, that takes ca. 12 hours to form (Péret et al., 2012 *Nat Cell Biol* 14:10, 991-8).

*Based on the current results the authors cannot really conclude that auxin coming from xylem pole pericycle cells (instead of divided pericycle cells or LRP) activates expression of *LAX3*. The authors should determine first the relationship between *LAX3* cortical expression and divided/undivided pericycle cells after D5-IAM application.*

***LAX3* expression is observed in front of LRP at stage I (Fig. 1C and D), suggesting that auxin moves from the pericycle cells very early on (before the first anticlinal division). Indeed, production of auxin in the pericycle using the J0121>>iaaH line triggers expression of *LAX3* in the cortex (Fig. 3G-L). This experiment demonstrate that auxin accumulating in the xylem pole pericycle cells moves towards the outer tissues. This observation is independent of whether pericycle cells have started dividing or not.**

4) *The title and abstract should be aimed for a broader audience.*

We agree with the reviewer and have simplified both accordingly.

5) *There are several concepts that need to be explained better and put in a developmental context (auxin, auxin-carrier, auxin transport...). Some other concepts should be used more precisely. For instance, the authors refer to 'the signal auxin' or 'the inductive auxin signal', which might be appropriate in a Physics paper, but in this paper it should be clearer that auxin is a plant hormone*

required for morphogenetic process and that it provides positional information. Mentioning 'LAX3 regulatory network' is also confusing as LAX3 is not a transcriptional regulator but an auxin carrier. Finally, the use of the words 'source of auxin' to refer to LRPs is misleading as it is not known if auxin is being produced in the LRPs or just passes through them -coming from vascular tissues.

We appreciate the reviewers point and, given the larger Article format of the current manuscript, have endeavoured to explain better several of the key terms and concepts.

6) *It is not clear why the models do not predict expression of LAX3 in the endodermis regardless of using LRPs in the model or not.*

Our model follows a number of rules that are based on experimental observations. As LAX3 expression has never been observed in the endodermis experimentally (Swarup et al, 2008 ; this manuscript), the model does not allow induction of LAX3 in this tissue.

7) *Fig S10, it is unclear why the addition of the percentages of all the bars in the graph is greater than 100%*

In Fig. S10 (now S6), black bars correspond to the number of LRP 18 hours after induction and the grey bars after 42 hours. The sum of black bars equal 100% and the sum of grey bars equal 100%. Both bars are shown on the same chart for facility of visualisation, as reported previously (Péret et al., 2012 Nat Cell Biol 14:10, 991-8 and Péret et al., 2012 Plant Cell 24 :2874-2885).

2nd Editorial Decision

17 May 2013

Thank you again for submitting your work to Molecular Systems Biology. First of all, I would like to apologize for the delay in getting back to you. Unfortunately it took a considerably long time until we found reviewers for evaluating the manuscript. We have now heard back from the three referees who agreed to evaluate your work. As you will see from the comments below, they provide rather contrasted views: reviewer #1 "does not believe this paper provides any scientific novelty nor further advances our understanding" whereas reviewer #2 feels that "This paper is a milestone for root biology in particular and plant biology in general". Therefore we had a closer look at the concerns raised by reviewer #1 (former reviewer #3).

Reviewer #1 is still concerned that the model relies on incorrect assumptions. The main issue raised by this reviewer is that pericycle (XPP) cells divide leading to a set of "12-18 cells in a stage I primordium" (as shown in transverse section in Fig. 1B, stage I) instead of the approximately 3 pericycle cells used in the model (as shown in the radial section Fig. 1A). This would then question the conclusions with regard to the mechanisms of localized induction of LAX3 expression in cortical cells. In our view, imposing the constraint of no LAX3 expression in endodermis is legitimate since it is an experimental fact, but it might be helpful to be more explicit on how auxin transport through the endodermis is modeled. In particular:

- The temporal scope of the model should be defined more explicitly: is it LRP at (early) stage I, prior of the division of pericycle cells?
- It is possible that some misunderstanding may have arisen from a discrepancy between the depiction of a radial section in Fig 1A, where XPP cells (purple) seem to be adjacent to each other and the transverse sections at stage I in Fig. 1B, where 5-6 cells are now already interspersed between the initial purple pericycle cells. It would help to indicate whether there is a difference in timing or indicate with a horizontal line in the panel 1B stage I the level of the radial section shown in Fig. 1A. Additionally, the XPP cells should be clearly indicated on Fig. 1C, as the meaning of the asterisk is somewhat unclear.
- More generally, the relationship between the drawings, the imaging data and the 3D model should be made more consistent and the information concerning the temporal stage and spatial (vertical) position should be more precise. This refers to the schemes in Fig. 1A and 1B, the exact temporal stages and spatial level of the fluorescent imaging data and the structure of the extruded transverse-section in the 3D model. This is particularly crucial when modeling and experimental imaging data are shown side-by-side (i.e. Figure 5).

- The need for and the value of a 3D model could be illustrated by 3D visualization, a staggered radial section visualization (or possibly a movie) in the Supplemental Info.

From the data shown in Fig. 3 and Fig. 5, it seems to us that the spatial arrangement assumed by the model corresponds to the architecture of the tissue at the stage of observation. Therefore we feel that you should be given the opportunity to respond to the critique raised by Reviewer #1 by replying to points listed above and by including the necessary clarifications. Furthermore, reviewer #3 raises several points regarding critical assumptions in the model, which should be convincingly addressed.

If you feel you can satisfactorily deal with these points and those listed by the referees, you may wish to submit a revised version of your manuscript. Please attach a point-by-point response letter giving details of the way in which you have handled each of the points raised by the referees. A revised manuscript will be once again subject to review and you probably understand that we can give you no guarantee at this stage that the eventual outcome will be favorable.

REFEREE REPORTS

Reviewer #1

The paper has been overall revised and the quality of the figures greatly increased. Unfortunately, the authors insist on neglecting (in this paper) the fact regular pericycle cells undergo cell division in order to become a stage I LR primordium. As known from previous studies and as shown in figure 1B of this paper, 2 consecutive pericycle cells (in purple) become 4-6 cells (in white); if this happens to 3 rows of pericycle cells as some of the authors recently published (Lucas et al, PNAS 2013 110 (13) 5229-5234), this means that there are a total of 12-18 cells in a stage I primordium versus 3 regular or non-divided pericycle cells. These newly formed cells are now constrained in the previous volume of the original pericycle cells. Subsequently, the morphology and geometry of a stage I primordium is obviously and undeniably different from regular pericycle cells, which is what the authors model. Previous studies from 16 years ago where the microscopical resolution was not nearly comparable to nowadays cannot be used to state that 'the changes in shape at Stage I are negligible as commonly reported in the literature (Malamy and Benfey, 1997). This is not a scientifically acceptable argument.

The major strength and novelty of this paper is the 3D modeling using 'morphologically realistic' cells. However, the authors build a model with the wrong cell morphologies and number: fewer number of cells and larger (pericycle cells) versus greater number of cells and smaller (stage I primordium). The use of a conceptually wrong model invalids any derived result, and in fact it calls into question if the model is just made to fit previous results they authors had rather than actually used to infer any result at all. In addition, they authors say in response to one question raised by this reviewer that: 'As LAX3 expression has never been observed in the endodermis experimentally (Swarup et al, 2008 ; this manuscript), the model does not allow induction of LAX3 in this tissue'. To what extent is the model built to fit what they authors expect to see?

In conclusion, I do not believe this paper provides any scientific novelty nor further advances our understanding of biological problems. Therefore I do not recommend its publication.

Reviewer #2

This paper is a milestone for root biology in particular and plant biology in general. The concisely and very well written paper aims to shed light into mechanisms that lead to emergence of lateral root primordia by using experimental knowledge about the regulatory components that control auxin inducible LAX3 expression in combination with mathematical modeling of the regulatory network controlling LAX3 induction with auxin movement. By doing so the author team was able not only to unravel important aspects of the mechanisms regulating LAX3, but through iterative modeling cycles and experimental action to predict a new regulatory component which turned out to be PIN3. Moreover, by integrating PIN3 into the model and by experimental verification a robust the model was generated in which auxin sink/source relation was causally linked to sequential induction of PIN3 and LAX3 being robust to variations in tissue geometry and magnitude of auxin source.

Congratulations! This paper excellently fits scope and format of msb and can be published as it is.

Reviewer #3

Peret et al have used a combination of confocal microscopy, molecular perturbations and mathematical modeling to analyse auxin transport and signalling in the context of lateral root initiation. The work is presented as an iterative process where model predictions are used to define new experiments. New hypotheses are added to the model in two steps where the final model is claimed to be the only one able to explain the data provided. The experiments provide novel insights into the lateral root initiation process, and the models provide information on how some of the added mechanisms leads to changes in LAX induction in cortical cells.

The experiments are showing new dynamics and the models are developed with aspect to robustness on changing local geometries, but the model evaluation and comparison lack some stringency, which may heavily affect the mechanistic interpretation of the added model hypotheses. Some major points are:

1) The model is using several assumptions that may influence the result although they are outside the focus of the paper. Still it is very hard as a reader to understand how much these factors influence the result. It would be very helpful to quantify/estimate the dependence on factors such as potential apoplastic transport (could this sharpen the number of cells in the cortex that get auxin from the source), potential intracellular auxin gradients, the unidentified efflux carrier polarization in the endodermis, the transport polarization in the XPP vs other pericycle cells, and the total lack of active auxin transporters in the epidermis. For the auxin transporter expressions/localizations, are there any indication from experiments?

2) More directly related to the focus of the paper, several of the authors are co-authors of another PIN3 story for lateral root initiation (Marhavy et al (2013), EMBO Journal 32, 149-158). How does the current work relate to the findings in this paper, and how would the described PIN3 dynamics in the endodermis affect the results?

3) To select one model over the other the arguing is "the model (1) could not capture the spatially restricted pLAX3:LAX3-YFP expression pattern". But this seems to be highly dependent on the source. If for example comparing Figs. M5 and M11 (source 2 root 2), my impression is that in both cases there is a source strength where a single cortical cell file is expressing LAX. Also, when the first model is discussed there is a discussion on how this model is heavily dependent on the auxin source, while this is not discussed for the final model. However no quantification of this robustness is presented, and the auxin source plots in the supplement do not directly convince this reviewer. A second selection criteria is the NOA experiment, described as an influx carrier malfunction perturbation. In the simulation active influx transport is set to 50% of its original value, but also the auxin source is increased a factor 20 (according to the supplemental text). How important is this increase in auxin source and how was the factor chosen? Would there be a factor leading to more cortical cells expressing LAX also in model 1? Given point (1) above, this then highlights the need of an objective and quantitative comparison of the different models, where the unknown parameters used in the models that are outside the main focus are 'investigated/optimized' equally for all model versions.

Other comments:

The model work is claimed to follow previous modeling efforts (refs Swarup 2005, Grieneisen 2007, Laskowski 2008), but all these have a subcellular compartmentalization where e.g. internal auxin gradients and apoplastic transport can be represented. My impression of the model presented in this paper is that it is cell-based with no apoplastic transport, and it is not clearly stated that it is quite different from the others.

There is a claim that multiscale modelling is used. The impression this reviewer gets is that, although a fine mesh was defined for cell geometries, in the end this was summed when calculating cell-cell interactions. The model has a genetic network per cell and auxin transport between (single) cell compartments (with volumes and areas accounted for). Do the authors claim that this represents

multiscale? Do all models including transport between cells represent multiscale models?

My impression was that when fitted to a Hill function, the LAX auxin response has an optimal Hill coefficient of two, while I get the impression that the value three has been used in the simulations. Is there a reason for this?

A model hypothesis is that there is an auxin-induced efflux carrier in the cortex cells, but such transporter will need to be localized at the membrane to change the result. Was the hypothesis that an efflux carrier was auxin-inducible in the cortex cells and polarized in a specific pattern? The impression is that the model used a polarization (later) measured for PIN3.

How was the NOA/NPA modeled? How was the transport adjusted and did the source change?

Minor comments:

line 242 from just from
line 253 radial direction -> circumferential?
line 255 Figure M3
line 549 (see)

2nd Revision - authors' response

03 July 2013

Reviewer #1

The paper has been overall revised and the quality of the figures greatly increased.

We thank reviewer #1 for these very positive comments.

Unfortunately, the authors insist on neglecting (in this paper) the fact regular pericycle cells undergo cell division in order to become a stage I LR primordium. As known from previous studies and as shown in figure 1B of this paper, 2 consecutive pericycle cells (in purple) become 4-6 cells (in white); if this happens to 3 rows of pericycle cells as some of the authors recently published (Lucas et al, PNAS 2013 110 (13) 5229-5234), this means that there are a total of 12-18 cells in a stage I primordium versus 3 regular or non-divided pericycle cells. These newly formed cells are now constrained in the previous volume of the original pericycle cells. Subsequently, the morphology and geometry of a stage I primordium is obviously and undeniably different from regular pericycle cells, which is what the authors model. Previous studies from 16 years ago where the microscopical resolution was not nearly comparable to nowadays cannot be used to state that 'the changes in shape at Stage I are negligible as commonly reported in the literature (Malamy and Benfey, 1997). This is not a scientifically acceptable argument.

Whilst the XPP cells (from which the LRP develop) do divide between stages zero and one, as noted by the reviewer, daughter cells are always contained within the original volume of the parent cells. Furthermore, the XPP daughter cells in the same file (up to stage one) express DR5 and likely act as a source of auxin (see this work and Benkova et al. 2003). Thus, we had originally reasoned that the cell divisions should not alter the overall nature of the auxin source from the perspective of the overlying tissues and hence the model behaviour. Nevertheless, as requested, we have now performed the necessary simulations and included them in the revised manuscript. However, as now noted in the main text, the inclusion of these cell divisions leads to indistinguishable differences in the model behaviour (see Supplementary Modelling Figure M2).

The major strength and novelty of this paper is the 3D modeling using 'morphologically realistic' cells. However, the authors build a model with the wrong cell morphologies and number: fewer number of cells and larger (pericycle cells) versus greater number of cells and smaller (stage I primordium). The use of a conceptually wrong model invalids any derived result, and in fact it calls into question if the model is just made to fit previous results they authors had rather than actually used to infer any result at all.

We are puzzled by the reviewers reasoning having clearly included 'morphologically realistic' cells

in our models. Furthermore, in the revised manuscript we have demonstrated that: 1) including these additional cell divisions does not impact the behaviour of the model, and 2) as demonstrated throughout the text, the model leads to useful mechanistic insights and experimentally verifiable predictions.

In addition, they authors say in response to one question raised by this reviewer that: 'As LAX3 expression has never been observed in the endodermis experimentally (Swarup et al, 2008 ; this manuscript), the model does not allow induction of LAX3 in this tissue'. To what extent is the model built to fit what they authors expect to see? In conclusion, I do not believe this paper provides any scientific novelty nor further advances our understanding of biological problems. Therefore I do not recommend its publication.

Once again, we are puzzled by the logic of the reviewer 1's reasoning. I am sure that reviewer 1 agrees that a systems biology model has to be built on a set of rules derived from experimental observations. Furthermore, as noted by reviewer 2 and clearly stated by reviewer 3 "*The work is presented as an iterative process where model predictions are used to define new experiments.*" Hence, we strongly disagree with reviewer 1 that there was any attempt to 'fit' our model to our experimental observations (and personally find this insinuation very insulting and unprofessional). We would like to note that the arguments made above contrast markedly with the positive comments and enthusiasm expressed by the other reviewers about the novelty and importance of the work.

Reviewer #2

This paper is a milestone for root biology in particular and plant biology in general. The concisely and very well written paper aims to shed light into mechanisms that lead to emergence of lateral root primordia by using experimental knowledge about the regulatory components that control auxin inducible LAX3 expression in combination with mathematical modeling of the regulatory network controlling LAX3 induction with auxin movement. By doing so the author team was able not only to unravel important aspects of the mechanisms regulating LAX3, but through iterative modeling cycles and experimental action to predict a new regulatory component which turned out to be PIN3. Moreover, by integrating PIN3 into the model and by experimental verification a robust the model was generated in which auxin sink/source relation was causally linked to sequential induction of PIN3 and LAX3 being robust to variations in tissue geometry and magnitude of auxin source. Congratulations! This paper excellently fits scope and format of msb and can be published as it is.

We are very grateful to reviewer #2 for pinpointing the novelty and merit of our work and its suitability for publication in MSB.

Reviewer #3

Peret et al have used a combination of confocal microscopy, molecular perturbations and mathematical modeling to analyse auxin transport and signalling in the context of lateral root initiation. The work is presented as an iterative process where model predictions are used to define new experiments. New hypotheses are added to the model in two steps where the final model is claimed to be the only one able to explain the data provided. The experiments provide novel insights into the lateral root initiation process, and the models provide information on how some of the added mechanisms leads to changes in LAX induction in cortical cells.

The experiments are showing new dynamics and the models are developed with aspect to robustness on changing local geometries, but the model evaluation and comparison lack some stringency, which may heavily affect the mechanistic interpretation of the added model hypotheses. Some major points are:

1) The model is using several assumptions that may influence the result although they are outside the focus of the paper. Still it is very hard as a reader to understand how much these factors influence the result. It would be very helpful to quantify/estimate the dependence on factors such as potential apoplasmic transport (could this sharpen the number of cells in the cortex that get auxin

from the source), potential intracellular auxin gradients, the unidentified efflux carrier polarization in the endodermis, the transport polarization in the XPP vs other pericycle cells, and the total lack of active auxin transporters in the epidermis. For the auxin transporter expressions/localizations, are there any indication from experiments?

We understand the reviewer's concerns and now discuss each of these issues in the main text. In particular, the role of apoplastic diffusion and its significance has already been discussed elsewhere in Kramer (2006), *Plant Physiology*, 144, no. 4 and Kramer (2007), *J. Exp. Bot.* 59, no. 1. In particular, movement of auxin in the apoplast is dominated by carrier-mediated transport, and furthermore we estimate (based on his calculations) that an auxin molecule (when LAX3 is expressed) will travel no more than 1 μ M. Since this is far less than a cortical or endodermal cell diameter (which is of the order of 10-15 μ M) we do not expect that this could lead to a sharpening effect or a significant alteration of the predicted pattern. Importantly (and as noted in the text) the estimated distances are independent of the rate at which the auxin enters the apoplast (and hence the magnitude of the auxin source).

Although the estimates for the diffusion rate of auxin inside cells is thought to be slow enough for it to form intracellular gradients, this is the case when cells are particularly long (over 100 μ M) and if carriers (notably PINs) are localized in a polar fashion to one end of the cell (i.e. so at one of the cell there is sink, mediated by PIN accumulation in that cell, and the other end is a source, provided by PIN in the abutting cell). However, carriers in our model are predominantly apolar, except PIN3 in the cortex, which localises on all cell walls, except those facing the endodermal cells. This asymmetry in PIN3 polarity could potentially lead to an apical to basal intracellular auxin gradient, but the distance between the apical and basal ends of a cortical cell is rather short (~15 μ M) and so it is unlikely that a significant concentration gradient would form in this direction.

Our assumptions regarding the movement of auxin from the pericycle cells to the cortex are rather general, in that the transporters allow the auxin to move in a predominantly radial direction from the pericycle to the cortex via the endodermal cells. We note that since auxin is trapped once it enters the cortex (and does not move back into the endodermis, due to the lack of the necessary efflux carriers), auxin in the endodermis can be thought of as an "input" pre-pattern which LAX3 in the cortex must then sharpen (i.e. the interaction between the endodermis and cortex is largely "one way"). So, one further assumption could be that the auxin can also move circumferentially or longitudinally, from endodermal to endodermal cell. However, this would simply widen the range of the auxin pre-pattern in the endodermis. This is unlikely in light of our experimental results, which demonstrates that auxin emanates from just one XPP cell file so that auxin accumulates in just two overlaying endodermal cells.

The lack of other auxin transporters reflects our observations for the particular stage of development we are considering (stage one). However, as now mentioned in the main text, PIN3 is likely to be also expressed in the epidermis at a later stage of development - but this is not known and a subject for future study. To our knowledge, only this work and the work of Marhavy et al (2012) provides information of the localization of the known auxin transporters in the outer layers (as noted in the main text, we performed an survey of all the known auxin efflux transporters in the outer layers in the mature root and only PIN3 expression was detected).

2) More directly related to the focus of the paper, several of the authors are co-authors of another PIN3 story for lateral root initiation (Marhavy et al (2013), EMBO Journal 32, 149-158). How does the current work relate to the findings in this paper, and how would the described PIN3 dynamics in the endodermis affect the results?

In the work of Marhavy et al (2013), PIN3 is polarly localised towards the XPP source cells and has a role in regulating the development of the LRP. As now noted in the main text and above (point 2), auxin accumulation in the endodermis can be thought of as a pre-pattern to which the cortical LAX3 responds. In their experiments, the expression of PIN3 in the endodermis (which is established very early on - prior to LAX3 expression) may alter the overall magnitude of auxin accumulation in the endodermal cells, it will not alter (for example) the number of endodermal cells in which auxin accumulates (and hence the spatial extent of the auxin input pattern provided by the endodermis). Thus, inclusion of PIN3 in the endodermis will not effect the model behaviour. Furthermore, we note that we could not detect PIN3 in the endodermis in our experiments. This could be because in

Marhavy et al (2013) the lateral roots were being mechanically induced, whereas in our case no external stimulation was applied (and mechanical stimulation is known to induce changes in other PIN family members - see Ditengou (2008).

3) To select one model over the other the arguing is "the model (1) could not capture the spatially restricted pLAX3:LAX3-YFP expression pattern". But this seems to be highly dependent on the source. If for example comparing Figs. M5 and M11 (source 2 root 2), my impression is that in both cases there is a source strength where a single cortical cell file is expressing LAX. Also, when the first model is discussed there is a discussion on how this model is heavily dependent on the auxin source, while this is not discussed for the final model. However no quantification of this robustness is presented, and the auxin source plots in the supplement do not directly convince this reviewer.

We thank the reviewer for raising this important point. In fact, we had already provided criteria for counting how many cortical cells express LAX3. This was provided in Materials and Methods, under the subheading Statistical Analysis, but we now state it more explicitly in the main text. In particular, we assume that the auxin supplied by the primordia must be high enough to (on average) induce LAX3 expression levels in at least one cell file to 50% - which is not the case in the example pointed out by the reviewer. We note that by "sensitivity" we were predominantly thinking of the number of cells that express LAX3, however we now recognise that what was originally written may have appeared rather vague and hence misleading. Nevertheless, in light of comment 3 below, we agree that a more robust analysis is required to reject model version one. Fortunately, we were able to take advantage of methods from asymptotic analysis and derive approximations to the steady-states of the full model. These approximate expressions are very accurate (see Supplementary Modelling Figure M4-M5, which are now placed at the end of the Supplementary Modelling Information file), but are now amenable to mathematical analysis. As described now in the main text (and in more detail in the Supplementary Modelling Information, Section 5), for model version one to be compatible with all our experimental data, we would require a much sharper steady-state LAX3 response to changes in auxin levels than observed experimentally (Figure 3H).

A second selection criteria is the NOA experiment, described as an influx carrier malfunction perturbation. In the simulation active influx transport is set to 50% of its original value, but also the auxin source is increased a factor 20 (according to the supplemental text). How important is this increase in auxin source and how was the factor chosen? Would there be a factor leading to more cortical cells expressing LAX also in model 1?

We would first like to point out that the 50% reduction in carrier activity is an error - we actually set the carrier activity to zero in the simulations shown. As now described in the main text, NOA affects influx carriers, but may have additional secondary effects such as increasing the level of auxin being supplied by the primordia (see main text for details). For this reason, in model version one, we tested whether the spread could be accounted for by increasing the source of the auxin supplied by the primordia. Given a factor 20 fold change, LAX3 would only be expressed in those cortical cells making indirect contact (via endodermal cells) with the XPP source cells. We also tested for even higher fold changes (200) - and in this case LAX3 expression in one additional cell file. This lack of a significant spread reflects the fact that auxin is effectively trapped inside cells (as a consequence of acid trapping - see Supplementary Modelling Information, Section 1.2 for further details). Factor 20 was chosen because it is a relatively large number. Since this represents levels of auxin 20 fold higher than the normal level required to induce LAX3 expression - we reasoned that even higher fold changes (i.e. of 100-200) would likely reflect un-physiological (and most likely toxic) auxin levels.

The spread of LAX3 in the NOA treatments indicated that there was some additional transport factor missing from the model. In particular, since LAX3 expression gradually, spreads from the cortical cells overlying the primordia to their neighbours, we reasoned (as now mentioned in the main text) that there was an auxin inducible efflux carrier polarised towards other neighbouring cortical cells. Whilst this results in a much greater spread than for model version one, it could not match the full extent of the spread observed experimentally. Thus, we also ran simulations where the source was again increased by a factor of 20. This time, unlike in model version one, LAX3 expression spread to all cortical cells.

Given point (1) above, this then highlights the need of an objective and quantitative comparison of the different models, where the unknown parameters used in the models that are outside the main focus are 'investigated/optimized' equally for all model versions.

As noted in point one, many of factors cited by the reviewer are likely to have negligible impact on the model results. Furthermore, we performed a mathematical analysis of model version one and demonstrated that, given the available data on the LAX3 response curve (Figure 3H), the model will be unable to account for the stereotypical LAX3 expression patterns. We are grateful to the reviewer for pointing out these issues as we believe the inclusion of this mathematical analysis greatly strengthen the paper.

Finally, we would like to take this opportunity to point out the key difference between model versions one and two. In model version one (as is apparent from the mathematical analysis) the lack of an efflux carrier means there is little communication between cortical cells. Hence (as noted above) they are simply providing direct readout of the concentration of auxin coming from the underlying endodermal cells, and for this readout to generate sharp patterns, the LAX3 response must be almost switch like (which is not the case).

In model version two, the inclusion of PIN3/AEC means that cortical cell can directly communicate with each other via auxin. As can be demonstrated by our simulations, this allows for strong intercellular gradients of LAX3 expression to form. However, what is important is that the shape of the predicted LAX3 response to exogenous auxin is the same regardless of whether PIN3 is expressed or not (Supplementary Modelling Figure M3). This explains the apparent discrepancy between the sharp LAX3 expression pattern and the non-switch like response given by LAX3 to exogenous auxin. In summary this is the reason why we reject model version one in favour of model version two.

Other comments:

The model work is claimed to follow previous modeling efforts (refs Swarup 2005, Grieneisen 2007, Laskowski 2008), but all these have a subcellular compartmentalization where e.g. internal auxin gradients and apoplastic transport can be represented. My impression of the model presented in this paper is that it is cell-based with no apoplastic transport, and it is not clearly stated that it is quite different from the others.

The reviewer is entirely correct and we have updated the text accordingly.

There is a claim that multiscale modelling is used. The impression this reviewer gets is that, although a fine mesh was defined for cell geometries, in the end this was summed when calculating cell-cell interactions. The model has a genetic network per cell and auxin transport between (single) cell compartments (with volumes and areas accounted for). Do the authors claim that this represents multiscale? Do all models including transport between cells represent multiscale models?

We agree that the model is better described as a cell based or multiple model, rather than multiscale one, as this often corresponds to cases where intracellular gradients (for example) are included.

My impression was that when fitted to a Hill function, the LAX auxin response has an optimal Hill coefficient of two, while I get the impression that the value three has been used in the simulations. Is there a reason for this?

The effective Hill coefficient, as experimentally measured, and the Hill coefficient corresponding to interactions between ARFs/Factor X on the LAX3 promoter are not necessarily equal. We have now included comments in the text to emphasize this. In particular, as now noted in Supplementary Modelling Information, Section 4.2, for $m=3$, the observable (or effective) Hill coefficient is ~ 2 , i.e. parameters have been chosen to fit with the data presented in Figure 3H.

A model hypothesis is that there is an auxin-induced efflux carrier in the cortex cells, but such transporter will need to be localized at the membrane to change the result. Was the hypothesis that an efflux carrier was auxin-inducible in the cortex cells and polarized in a specific pattern? The impression is that the model used a polarization (later) measured for PIN3.

As now described in the text, we initially assumed an auxin efflux carrier was polarized towards neighbouring cortical cells.

How was the NOA/NPA modeled? How was the transport adjusted and did the source change?

We apologise for not including this information in the submitted version of the manuscript. As now described in the Supplementary Modelling Information, the NOA/NPA experiment was simulated by setting the permeability of PIN3/LAX3 to zero. The response provided by the model under these conditions - for various level auxin source - are provided in Supplementary Modelling Figures M20-M25 (these were provided with the original submission but we neglected to reference them in the main text). Since in this case the auxin is effectively trapped once it enters the cortex (since we have removed all influx/efflux carriers) the results will be valid for a very wide range of auxin sources (see comments regarding the NOA experiment above).

Minor comments:

line 242 from just from

line 253 radial direction -> circumferential?

line 255 Figure M3

line 549 (see)

These modifications have been taken into account.